# Federated Compositional Optimization: The Impact of Two-Sided Learning Rates on Communication Efficiency

## Abstract

Compositional optimization (CO) has recently gained popularity due to its applications in distributionally robust optimization (DRO), meta-learning, reinforcement learning, and many other machine learning applications. The large-scale and distributed nature of data necessitates efficient federated learning (FL) algorithms for CO, but the compositional structure of the objective poses significant challenges. Current methods either rely on large batch gradients (which are impractical) or suffer from suboptimal communication efficiency. To address these challenges, we propose efficient FedAvg-type algorithms for solving non-convex CO in the FL setting. We first establish that standard FedAvg fails in solving the federated CO problems due to data heterogeneity, which amplifies bias in local gradient estimates. Our analysis establishes that either *additional communication* or *two-sided learning rate-based* algorithms are required to control this bias. To this end, we develop two algorithms for solving the federated CO problem. First, we propose FedDRO that utilizes the compositional problem structure to design a communication strategy that allows FedAvg to control the bias in the estimation of the compositional gradient, achieving $\mathcal{O}(\epsilon^{-2})$ sample and $\mathcal{O}(\epsilon^{-3/2})$ communication complexity. Then we propose DS-FedDRO, a two-sided learning rate algorithm, that eliminates the need for additional communication and achieves the optimal $\mathcal{O}(\epsilon^{-2})$ sample and $\mathcal{O}(\epsilon^{-1})$ communication complexity, highlighting the importance of two-sided learning rate algorithms for solving federated CO problems. Both algorithms avoid the need for large batch gradients and achieve linear speedup with the number of clients. We corroborate our theoretical findings with empirical studies on large-scale DRO problems.

## 1 Introduction

Compositional optimization (CO) problems deal with the minimization of the composition of functions. A standard CO problem takes the form

$$\min_{x \in \mathbb{R}^d} f(g(x)) \quad \text{with} \quad g(x) \coloneqq \mathbb{E}_{\zeta \sim \mathcal{D}_g}[g(x; \zeta)], \tag{1}$$

where $x \in \mathbb{R}^d$ is the optimization variable, $f : \mathbb{R}^{d_g} \to \mathbb{R}$ and $g : \mathbb{R}^d \to \mathbb{R}^{d_g}$ are smooth functions, and $\zeta \sim \mathcal{D}_g$ represents a stochastic sample of $g(\cdot)$ from distribution $\mathcal{D}_g$. CO finds applications in a broad range of machine learning applications, including but not limited to distributionally robust optimization (DRO) Qi et al. (2022), meta-learning Finn et al. (2017), phase retrieval Duchi & Ruan (2019), portfolio optimization Shapiro et al. (2021), and reinforcement learning Wang et al. (2017).

In this work, we focus on a more challenging version of the CO problem (1) that often arises in the DRO formulation Haddadpour et al. (2022). Specifically, the problems that jointly minimize the summation of a compositional and a non-compositional objective. DRO has recently garnered significant attention because of its capability of handling noisy labels Chen et al. (2022), training fair machine learning models Qi et al. (2022), imbalanced Qi et al. (2020a) and adversarial data Chen & Paschalidis (2018). A standard approach to solve DRO is to utilize primal-dual algorithms Nemirovski et al. (2009) that are inherently slow because of a large number of stochastic constraints. The CO formulation enables the development of faster primal-only DRO algorithms Haddadpour

et al. (2022). The majority of existing works to solve CO problems consider a centralized setting, however, modern large-scale machine-learning applications are characterized by the distributed collection of data by multiple clients Kairouz et al. (2021). This necessitates the development of distributed algorithms to solve the DRO problem.

Federated learning (FL) is a distributed learning paradigm that allows clients to solve a joint problem in collaboration with a server while keeping the data of each client private McMahan et al. (2017). The clients act as computing units while the server orchestrates the parameter sharing among clients. Numerous FL algorithms exist in the literature to tackle standard (non-compositional) problems Li et al. (2019; 2020); Karimireddy et al. (2019); Sharma et al. (2019); Zhang et al. (2021); Khanduri et al. (2021). However, there is a lack of efficient distributed implementations when it comes to CO problems. The major challenges in developing FL algorithms for solving the CO problem are:

[**C1**]: Compositional structure of the problem leads to *biased* stochastic gradient estimates and this bias is amplified during local updates, which makes the analysis intractable Chen et al. (2021).

[**C2**]: Typically, data distribution at each client is different, referred to as data heterogeneity. Heterogeneously distributed compositional objective results in *client drift* during local updates that lead to divergence of federated CO algorithms. This is in sharp contrast to the standard FedAvg for non-CO objectives where client drift can be controlled during the local updates Karimireddy et al. (2019).

[**C3**]: A majority of algorithms for solving CO rely on accuracy-dependent large batch gradients where the batch size depends on the desired solution accuracy, which is not practical from an implementation point of view Huang et al. (2021); Haddadpour et al. (2022); Guo et al. (2022).

These challenges naturally lead to the following question:

*Can we develop FL algorithms that tackle* [**C1**] − [**C3**] *to solve CO in a FL setting?*

In this work, we address the above question and develop novel FL algorithms to solve CO problems. Although, our development focuses on the DRO problem the algorithms developed in our work have wider applicability to other general CO problems. The major contributions of our work are:

• We for the first time present a negative result that establishes that the vanilla FedAvg (customized to CO) is **incapable of solving** the CO problems as it leads to bias amplification during the local updates. This shows that either *additional communication/processing* or *non-classical aggregation* procedure is required by FedAvg to mitigate the bias in the local gradient estimation.

• We develop two novel FL algorithms FedDRO and DS-FedDRO, for solving problems with both **compositional and non-compositional non-convex objectives**. To our knowledge, such algorithms have been absent from the open literature so far. Importantly, FedDRO and DS-FedDRO address the above-mentioned challenges by developing several key innovations in the algorithm design.

  – FedDRO addresses [**C1**] by designing a **communication strategy** that utilizes the specific CO problem structure and allows us to control the gradient bias at the cost of additional low-dimensional communication. On the contrary, DS-FedDRO tackles [**C1**] by designing **2-sided learning rate** CO algorithms for FL wherein the server aggregation is performed similarly as the local updates.

  – To address [**C2**], we **design the local updates** at each client so that the client drift is bounded. Our analysis captures the effect of data heterogeneity on the performance of the algorithms.

  – To address [**C3**], we utilize a **hybrid momentum-based estimator** to learn the compositional embedding and combine it with a stochastic gradient (SG) estimator to conduct the local updates. This construction allows us to circumvent the need to compute large accuracy-dependent batch sizes for computing the gradients and the compositional function evaluations.

• We establish the **convergence** of FedDRO and DS-FedDRO and show that to achieve an $\epsilon$-stationary point both algorithms require $\mathcal{O}(\epsilon^{-2})$ samples while achieving **linear speed-up** with the number of clients, i.e., requiring $\mathcal{O}(K^{-1}\epsilon^{-2})$ samples per client. Moreover, FedDRO achieves a communication complexity of $\mathcal{O}(\epsilon^{-3/2})$ while DS-FedDRO achieves $\mathcal{O}(\epsilon^{-1})$.

## 2 PROBLEM

In this work, we focus on a general version of the CO problem defined in (1). We consider the following problem that often arises in DRO (see Section 2.1) in a distributed setting with $K$ clients

$$\inf_{x \in \mathbb{R}^d} \left\{ \Phi(x) := h(x) + f(g(x)) \right\} \text{ with } h(x) := \frac{1}{K} \sum_{k=1}^{K} h_k(x) \ \& \ g(x) := \frac{1}{K} \sum_{k=1}^{K} g_k(x), \quad (2)$$

where each client $k \in [K]$ has access to the local functions $h_k : \mathbb{R}^d \to \mathbb{R}$ and $g_k : \mathbb{R}^d \to \mathbb{R}^{d_g}$ while $f(\cdot)$ is same as (1). The local functions $h_k(\cdot)$ and $g_k(\cdot)$ at each client $k \in [K]$ are: $h_k(x) = \mathbb{E}_{\xi_k \sim \mathcal{D}_{h_k}}[h_k(x; \xi_k)]$ and $g_k(x) = \mathbb{E}_{\zeta_k \sim \mathcal{D}_{g_k}}[g_k(x; \zeta_k)]$ and where $\xi_k \sim \mathcal{D}_{h_k}$ (resp. $\zeta_k \sim \mathcal{D}_{g_k}$) represents a sample of $h_k(\cdot)$ (resp. $g_k(\cdot)$) from distribution $\mathcal{D}_{h_k}$ (resp. $\mathcal{D}_{g_k}$). Moreover, the data at each client is heterogeneous, i.e., $\mathcal{D}_{h_k} \neq \mathcal{D}_{h_\ell}$ and $\mathcal{D}_{g_k} \neq \mathcal{D}_{g_\ell}$ for $k \neq \ell$ and $k, \ell \in [K]$.

In comparison to the basic CO in (1), (2) is significantly challenging, first, because of the presence of both compositional and non-compositional objectives and second, because of the distributed nature of the compositional function $g(\cdot)$. We would also like to point out that the algorithms and the analysis presented in this work can be easily extended to the problems where $f(\cdot) := 1/K \sum_{k=1}^{K} f_k(\cdot)$ is also distributed among $K$ agents with each agent having access to $f_k(\cdot)$ for $k \in [K]$.

*Remark* 2.1 (Comparison to Gao et al. (2022) and Huang et al. (2021)). Note that formulation (2) is significantly different than the setting considered in Huang et al. (2021); Gao et al. (2022). Specifically, our formulation considers a setting where the compositional functions are distributed across agents, i.e., the function is $g = 1/K \sum_{k=1}^{K} g_k(x)$. In contrast, Huang et al. (2021); Gao et al. (2022) consider a setting with objective $1/K \sum_{k=1}^{K} f_k(g_k(\cdot))$, note here that the compositional function is local to each agent. This implies that algorithms developed in Huang et al. (2021); Gao et al. (2022) cannot solve problem (2). Importantly, problem (2) models realistic FL training settings while being more challenging compared to Huang et al. (2021); Gao et al. (2022) since in (2) the data heterogeneity of the inner problem also plays a role in the convergence of the FL algorithm. Please see the discussion in Appendix A.1 for more details.

## 2.1 EXAMPLES: CO REFORMULATION OF DRO PROBLEMS

In this section, we discuss different DRO formulations that can be efficiently solved using CO Haddadpour et al. (2022). DRO problem with a set of $m$ training samples denoted as $\{\zeta_i\}_{i=1}^{m}$ is

$$\min_{x \in \mathbb{R}^d} \max_{\mathbf{p} \in P_m} \sum_{i=1}^{m} p_i \ell(x; \zeta_i) - \lambda D_*(\mathbf{p}, \mathbf{1}/m) \tag{3}$$

where $x \in \mathbb{R}^d$ is the model parameter, $P_m := \{\mathbf{p} \in \mathbb{R}^m : \sum_{i=1}^{m} p_i = 1, p_i \geq 0\}$ is $m$-dimensional simplex, $D_*(\mathbf{p}, \mathbf{1}/m)$ is a divergence metric that measures distance between $\mathbf{p}$ and uniform probability $\mathbf{1}/m \in \mathbb{R}^m$, and $\ell(x, \zeta_i)$ denotes the loss on sample $\zeta_i$, $\rho$ is a constraint parameter, and $\lambda$ is a hyperparameter. Next, we discuss two popular reformulations of (3) in the form of CO problems.

**DRO with KL-Divergence.** Problem (3) is referred to as a KL-regularized DRO when the distance metric $D_*(\mathbf{p}, \mathbf{1}/m)$ is the KL-Divergence, i.e., we have $D_*(\mathbf{p}, \mathbf{1}/m) = D_{\mathrm{KL}}(\mathbf{p}, \mathbf{1}/m)$ with $D_{\mathrm{KL}}(\mathbf{p}, \mathbf{1}/m) := \sum_{i=1}^{m} p_i \log(p_i m)$. For this case, an equivalent reformulation of (3) is

$$\min_{x \in \mathbb{R}^d} \log\left(\frac{1}{m} \sum_{i=1}^{m} \exp\left(\frac{\ell(x; \zeta_i)}{\lambda}\right)\right), \tag{4}$$

which is a CO with $g(x) = 1/m \sum_{i=1}^{m} \exp(\ell(x; \zeta_i)/\lambda)$, $f(g(x)) = \log(g(x))$ and $h(x) = 0$.

**DRO with $\chi^2$- Divergence.** Similar to KL-regularized DRO, (3) is referred to as a $\chi^2$-regularized DRO when $D_*(\mathbf{p}, \mathbf{1}/m)$ is the $\chi^2$-Divergence, i.e., we have $D_*(\mathbf{p}, \mathbf{1}/m) = D_{\chi^2}(\mathbf{p}, \mathbf{1}/m)$ with $D_{\chi^2}(\mathbf{p}, \mathbf{1}/m) := m/2 \sum_{i=1}^{m} (p_i - 1/m)^2$. For this case, an equivalent reformulation of (3) is

$$\min_{x \in \mathbb{R}^d} -\frac{1}{2\lambda m} \sum_{i=1}^{m} \left(\ell(x; \zeta_i)\right)^2 + \frac{1}{2\lambda}\left(\frac{1}{m} \sum_{i=1}^{m} \ell(x; \zeta_i)\right)^2 \tag{5}$$

with $g(x) = 1/m \sum_{i=1}^{m} \ell(x; \zeta_i)$, $f(g(x)) = g(x)^2/2\lambda$ and $h(x) = -\frac{1}{2\lambda m} \sum_{i=1}^{m} \left(\ell(x; \zeta_i)\right)^2$.

Note that both (4) and (5) can be equivalently restated in the FL setting of (2). Moreover, in addition to the DRO problems the developed algorithms can be used to solve general CO problems.

**Related work.** Please see Table 1 for a comparison of current approaches to solve CO problems in distributed settings. For a detailed review of centralized and distributed non-convex CO and DRO problems, please see Appendix A. Here, we point out some drawbacks of the current approaches to solving federated CO problems:

– None of the current works guarantee linear speedup with the number of clients Huang et al. (2021); Haddadpour et al. (2022); Tarzanagh et al. (2022); Gao et al. (2022).

– Utilize complicated multi-loop algorithms with momentum or VR-based updates Tarzanagh et al. (2022) that sometime require computation of large batch size gradients Haddadpour et al. (2022) to guarantee convergence. Such algorithms are not preferred in practical implementations.

Table 1: Comparison with the existing works. Here, CO-ND refers to CO with a non-distributed compositional part (see Remark 2.1). CO + Non-CO refers to problems with both CO and Non-CO objectives. VR refers to variance reduction. (I) and (O) refers to the inner and outer loop, respectively.
* Theoretical guarantees for GCIVR exist only for the finite sample setting with $m$ total network-wide samples.

| ALGORITHM | SETTING | UPDATE | BATCH | COMP. | COMM. |
|---|---|---|---|---|---|
| ComFedL Huang et al. (2021) | CO-ND | SGD | $\mathcal{O}(\epsilon^{-2})$ | $\mathcal{O}(\epsilon^{-4})$ | $\mathcal{O}(\epsilon^{-1})$ |
| Local-SCGDM Gao et al. (2022) | CO-ND | Momentum SGD | $\mathcal{O}(1)$ | $\mathcal{O}(\epsilon^{-2})$ | $\mathcal{O}(\epsilon^{-1})$ |
| FedNest Tarzanagh et al. (2022) | Bilevel | VR | $\mathcal{O}(1)$ | $\mathcal{O}(\epsilon^{-2})$ | $\mathcal{O}(\epsilon^{-1})$ |
| FedBiO Li et al. (2024) | Bilevel | VR | $\mathcal{O}(1)$ | $\mathcal{O}(K^{-1}\epsilon^{-2.5})$ | $\mathcal{O}(\epsilon^{-1.5})$ |
| FedMBO Huang et al. (2023) | Bilevel | SGD | $\mathcal{O}(\ln(\epsilon^{-1}))$ | $\mathcal{O}(K^{-1}\epsilon^{-2})$ | $\mathcal{O}(\epsilon^{-2})$ |
| SimFBO Yang et al. (2024) | Bilevel | SGD | $\mathcal{O}(1)$ | $\mathcal{O}(\epsilon^{-2})$ | $\mathcal{O}(\epsilon^{-1})$ |
| GCIVR* Haddadpour et al. (2022) | CO + Non-CO | VR | $\sqrt{m}$ (I), $m$ (O) | $\mathcal{O}(\sqrt{m}\epsilon^{-1} \wedge \epsilon^{-1.5})$ | $\mathcal{O}(\epsilon^{-1})$ |
| FedDRO (Ours) | CO + Non-CO | SGD | $\mathcal{O}(1)$ | $\mathcal{O}(K^{-1}\epsilon^{-2})$ | $\mathcal{O}(\epsilon^{-1.5})$ |
| DS-FedDRO (Ours) | CO + Non-CO | SGD | $\mathcal{O}(1)$ | $\mathcal{O}(K^{-1}\epsilon^{-2})$ | $\mathcal{O}(\epsilon^{-1})$ |

- Recently developed bilevel algorithms although in theory can be used to solve CO problems Tarzanagh et al. (2022); Li et al. (2024); Huang et al. (2023); Yang et al. (2024), however, since the algorithms are designed for bilevel problems they often have complicated structure, suffer from worse performance, and require sharing of additional parameters.

- Consider a restricted setting where the compositional objective is not distributed among nodes Huang et al. (2021); Gao et al. (2022). Importantly, the algorithms developed therein cannot solve the problem considered in our work (see Appendix A.1).

Our work addresses all these issues and develops, FedDRO, the first simple SGD-based FL algorithm to tackle CO problems with the distributed compositional objective. Please see Table 1 for a comparison of the above works.

## 3 PRELIMINARIES

In this section, we introduce the assumptions, definitions, and preliminary lemmas.

**Definition 3.1** (Lipschitzness). For all $x_1, x_2 \in \mathbb{R}^d$, a differentiable function $\Phi : \mathbb{R}^d \to \mathbb{R}$ is: Lipschitz smooth if $\|\nabla\Phi(x_1) - \nabla\Phi(x_2)\| \leq L_\Phi\|x_1 - x_2\|$ for some $L_\Phi > 0$; Lipschitz if $\|\Phi(x_1) - \Phi(x_2)\| \leq B_\Phi\|x_1 - x_2\|$ for some $B_\Phi > 0$ and; Mean-Squared Lipschitz if $\mathbb{E}_\xi\|\Phi(x_1;\xi) - \Phi(x_2;\xi)\|^2 \leq B_\Phi^2\|x_1 - x_2\|^2$ for some $B_\Phi > 0$.

We make the following assumptions on the local and global functions in the problem (2).

**Assumption 3.2** (Lipschitzness). The following holds

1. The functions $f(\cdot)$, $h_k(\cdot)$, $g_k(\cdot)$ for all $k \in [K]$ are differentiable and Lipschitz-smooth with constants $L_f, L_h, L_g > 0$, respectively.

2. The function $f(\cdot)$ and $h_k(\cdot)$ are Lipschitz with constants $B_f > 0$ and $B_h > 0$, respectively, and $g_k(\cdot)$ is mean-squared Lipschitz for all $k \in [K]$ with constant $B_g > 0$.

**Assumption 3.3** (Unbiased Gradient and Bounded Variance). The stochastic gradients and function evaluations of the local functions at each client are unbiased and have bounded variance, i.e.,

$$\mathbb{E}_{\xi_k}[\nabla h_k(x;\xi_k)] = \nabla h_k(x), \ \mathbb{E}_{\zeta_k}[\nabla g_k(x;\zeta_k)] = \nabla g_k(x), \ \mathbb{E}_{\zeta_k}[g_k(x;\zeta_k)] = g_k(x),$$
$$\mathbb{E}_{\zeta_k}[\nabla g_k(x;\zeta_k)\nabla f(y)] = \nabla g_k(x)\nabla f(y)$$

and
$$\mathbb{E}_{\xi_k}\|\nabla h_k(x;\xi_k) - \nabla h_k(x)\|^2 \leq \sigma_h^2,$$
$$\mathbb{E}_{\zeta_k}\|\nabla g_k(x;\zeta_k) - \nabla g_k(x)\|^2 \leq \sigma_g^2, \ \mathbb{E}_{\zeta_k}\|g_k(x;\zeta_k) - g_k(x)\|^2 \leq \sigma_g^2,$$

for some $\sigma_h, \sigma_g > 0$ and for all $x \in \mathbb{R}^d$ and $k \in [K]$.

**Assumption 3.4** (Bounded Heterogeneity). The heterogeneity $h_k(\cdot)$ and $g_k(\cdot)$ is characterized as

$$\sup_{x \in \mathbb{R}^d} \|\nabla h_k(x) - \nabla h(x)\|^2 \leq \Delta_h^2 \ \text{ and } \ \sup_{x \in \mathbb{R}^d} \|\nabla g_k(x) - \nabla g(x)\|^2 \leq \Delta_g^2,$$

for some $\Delta_h, \Delta_g > 0$ for all $k \in [K]$.

The above assumptions are commonplace in the context of non-convex CO problems. Specifically, Assumption 3.2 is required to establish Lipschitz smoothness of the $\Phi(\cdot)$ (see Lemma 3.5) and

is standard in the analyses of CO problems Wang et al. (2017); Chen et al. (2021). Assumption 3.3 captures the effect of stochasticity in the gradient/function evaluations while Assumption 3.4 characterizes the data heterogeneity among clients. We note that these assumptions are standard and have been utilized in the past to establish the convergence of many FL non-CO algorithms Yu et al. (2019a); Karimireddy et al. (2019); Zhang et al. (2021); Woodworth et al. (2020).

**Lemma 3.5** (Lipschitzness of $\Phi$). *Under Assumption 3.2 the compositional function, $\Phi(\cdot)$, defined in (2) is Lipschitz smooth with constant: $L_\Phi := L_h + B_f L_g + B_g^2 L_f > 0$.*

Lemma 3.5 establishes Lipschitz smoothness (Definition 3.1) of the compositional function $\Phi(\cdot)$. In general, $\Phi(\cdot)$ is a non-convex, and therefore, we cannot expect to globally solve (2). We instead rely on finding approximate stationary points of $\Phi(\cdot)$ defined next.

**Definition 3.6** ($\epsilon$-stationary point). A point $x$ generated by an algorithm is an $\epsilon$-stationary point of $\Phi(\cdot)$ if $\mathbb{E}\|\nabla\Phi(x)\|^2 \le \epsilon$, where the expectation is taken w.r.t. the stochasticity of the algorithm.

**Definition 3.7** (Sample and Communication Complexity). The sample complexity is the total (stochastic) gradient and function evaluations required to achieve an $\epsilon$-stationary solution. Similarly, communication complexity is the total communication rounds between the clients and the server required to achieve an $\epsilon$-stationary solution.

## 4 FEDERATED NON-CONVEX CO ALGORITHMS

In this section, we first establish the incapability of vanilla FedAvg to solve CO problems. Then, we design communication-efficient FL algorithms to solve the non-convex CO problem.

### 4.1 CANDIDATE FEDAVG ALGORITHMS

---
**Algorithm 1** Vanilla FedAvg for non-convex CO

---
1: **Input**: Parameters: $\{\eta^t\}_{t=0}^{T-1}$, $I$
2: **Initialize**: $x_k^0 = \bar{x}^0$, $y_k^0 = \bar{y}^0$
3: **for** $t = 0$ to $T - 1$ **do**
4:     **for** $k = 1$ to $K$ **do**
5:       Update:
$\begin{cases} \text{Compute } \nabla\Phi_k(x_k^t) \text{ using (6)} \\ x_k^{t+1} = x_k^t - \eta^t \nabla\Phi_k(x_k^t) \\ y_k^{t+1} = g_k(x_k^{t+1}) \end{cases}$
6:       **if** $t + 1 \mod I = 0$ **then**
7:         [Case 1] Share: $\begin{cases} x_k^{t+1} = \bar{x}^{t+1} \end{cases}$

          [Case 2] Share: $\begin{cases} x_k^{t+1} = \bar{x}^{t+1} \\ y_k^{t+1} = g_k(\bar{x}^{t+1}) \\ y_k^{t+1} = \bar{y}^{t+1} \end{cases}$
8:       **end if**
9:     **end for**
10: **end for**

---

In this section, we show that vanilla FedAvg is not suitable for solving federated CO problems of form (2). To establish this, we consider a deterministic setting with $h(x) = 0$. For this setting, the local gradients of $\Phi(\cdot)$ are estimated as

$$\nabla\Phi_k(x) = \nabla g_k(x_k)\nabla f(y_k), \quad (6)$$

where the sequence $y_k$ represents the local estimate of the inner function $g(x)$. To solve the above problem in a federated setup, we consider two candidate versions of FedAvg described in Case I and II of Algorithm 1. Similar to vanilla FedAvg, each agent performs multiple local updates within each communication round (see Step 5 of Algorithm 1). Moreover, since $g(x) := 1/k \sum_{k=1}^{k} g_k(x)$ with each agent $k \in [K]$ having access to only the local copy $g_k(\cdot)$, estimating $g(\cdot)$ locally within each communication round is not feasible. Therefore, each agent utilizes $y_k = g_k(x)$ as the local estimate of the inner function $g(\cdot)$. For communication, we consider two protocols. In the first setting, after $I$ local updates, in each communication round the agents share the locally updated parameters with the server and receive the aggregated parameter (see Case I in Step 7). In the second setting, in addition to the locally updated parameters the agents also share their local function evaluations $y_k^t = g_k(x_k^t)$ with the server and receive the aggregated embedding $\bar{y}^t$. This step is utilized to improve the local estimates of $g(\cdot)$ (see Case II in Step 7). The algorithm executes for a total of $\lfloor T/I \rfloor$ communication rounds.

Next, we show that Algorithm 1 is not a good choice to solve the federated CO problems.

**Theorem 4.1** (Vanilla FedAvg: Non-Convergence for CO). *There exist functions $f(\cdot)$ and $g_k(\cdot)$ for $k \in [K]$ satisfying Assumptions 3.2, 3.3, and 3.4, and an initialization strategy such that for a fixed number of local updates $I > 1$, and for any $0 < \eta^t < C_\eta$ for $t \in \{0, 1, \ldots, T - 1\}$ where $C_\eta > 0$ is a constant, the iterates generated by Algorithm 1 under both Cases I and II do not converge to the stationary point of $\Phi(\cdot)$, where $\Phi(\cdot)$ is defined in (2) with $h(x) = 0$.*

---

**Algorithm 2** Federated non-convex CO algorithm: FedDRO

1: **Input**: Parameters: $\{\beta^t\}_{t=0}^{T-1}, \{\eta^t\}_{t=0}^{T-1}, I$
2: **Initialize**: $x_k^{-1} = x_k^0 = \bar{x}^0, y_k^0 = \bar{y}^0$
3: **for** $t = 0$ to $T - 1$ **do**
4:     **for** $k = 1$ to $K$ **do**
5:       Local Update and Sharing: $\begin{cases} \text{Compute } \nabla\Phi_k(x_k^t; \bar{\xi}_k^t) \text{ using (7)} \\ x_k^{t+1} = x_k^t - \eta^t \nabla\Phi_k(x_k^t; \bar{\xi}_k^t) \\ \text{Compute } y_k^{t+1} \text{ using (8) and share with the server} \\ \text{Receive } \bar{y}^{t+1} \text{ from the server and update } y_k^{t+1} = \bar{y}^{t+1} \end{cases}$
6:       **if** $t + 1 \mod I = 0$ **then**
7:         Aggregation at Server: $\begin{cases} x_k^{t+1} = \bar{x}^{t+1} \end{cases}$
8:       **end if**
9:     **end for**
10: **end for**
11: **Return**: $\bar{x}^{a(T)}$ where $a(T) \sim \mathcal{U}\{1, ..., T\}$.

---

Theorem 4.1 establishes that vanilla FedAvg is not suitable for solving federated CO problems. This naturally leads to the question of how can we modify FedAvg such that it can efficiently solve CO problems of the form (2)? Theorem 4.1 suggests that sharing $y_k$'s in each iteration or a different server aggregation strategy is required to ensure convergence of FedAvg since sharing the iterates $y_k$'s only intermittently or simple averaging at the server leads to non-convergence of FedAvg. To this end, we propose to modify the FedAvg algorithm as presented in Algorithm 1 in two ways: 1) by sharing $y_k$ in each iteration $t \in \{0, 1, \ldots, T - 1\}$, and 2) by modifying the classical FedAvg aggregation to a 2-sided update where server updates $x$ and $y$ incrementally Reddi et al. (2020). The next result shows that the modified FedAvg using point 1 above resolves the non-convergence issue of FedAvg for solving CO problems.

**Theorem 4.2** (Modified FedAvg: Convergence for CO). *Suppose we modify Algorithm 1 such that $y_k^t = \bar{y}^t$ is updated at each iteration $t \in \{0, 1, \ldots, T - 1\}$ instead of $[t + 1 \mod I]$ iterations as in current version of Algorithm 1. Then if functions $f(\cdot)$ and $g_k(x)$ for $k \in [K]$ satisfy Assumptions 3.2, 3.3, and 3.4 such that for a fixed number of local updates $1 \le I \le \mathcal{O}(T^{1/4})$, there exists a choice of $\eta^t > 0$ for $t \in \{0, 1, \ldots, T - 1\}$ such that the iterates generated by (modified) Algorithm 1 converge to the stationary point of $\Phi(\cdot)$, where $\Phi(\cdot)$ is defined in (2) with $h(x) = 0$.*

Motivated by Theorem 4.2, we next develop a federated algorithm, FedDRO, to solve problem (2) in a general stochastic setting with $h(x) \ne 0$. Later, in Section 5 we develop DS-FedDRO and establish that the additional communication required by FedDRO can be avoided by utilizing a 2-sided learning rate algorithm and an additional heterogeneity assumption.

## 4.2 FEDDRO: FEDERATED NON-CONVEX CO ALGORITHM

In this section, we propose a novel distributed non-convex CO algorithm, FedDRO, for solving (2). Motivated by Theorem 4.2 above, we first develop a novel approach where the estimates of low-dimensional embedding $g(\cdot)$ are aggregated in each iteration while the high-dimensional model parameters are shared intermittently. Recall that for many practical problems (see Section 2.1 for DRO) the embedding $g(\cdot)$ is low-dimensional (e.g., $d_g = 1$), therefore, sharing of $g(\cdot)$ will be relatively cheap in contrast to the high-dimensional model parameters of size $d$ which can be very large and take values in millions or even in billions for modern overparameterized neural networks Vaswani et al. (2017). Moreover, to solve the CO problems for DRO the developed algorithms generally utilize batch sizes (for gradient/function evaluation) that are dependent on the solution accuracy Huang et al. (2021); Haddadpour et al. (2022). However, this is not feasible in most practical settings. In addition, to control the bias and to circumvent the need to compute large batch gradients, we utilize a momentum-based estimator to learn the compositional function (see (8)) Chen et al. (2021). This construction allows us to develop FedAvg-type algorithms for solving non-convex CO problems wherein the local updates resemble the standard SGD updates.

The detailed steps of FedDRO are listed in Algorithm 2. During the local updates each client $k \in [K]$ updates its local model $x_k^t$ for all $t \in [T]$ using the local estimate of the stochastic gradients in Step 6. The local stochastic gradient estimates for each client $k \in [K]$ are denoted by $\nabla\Phi_k(x_k^t; \bar{\xi}_k)$

and are evaluated using the chain rule of differentiation as

$$\nabla \Phi_k(x_k^t; \bar{\xi}_k^t) = \nabla h_k(x_k^t; \xi_k^t) + \nabla g_k(x_k^t; \zeta_k^t) \nabla f(\bar{y}^t) \tag{7}$$

where $\bar{\xi}_k^t = \{\xi_k^t, \zeta_k^t\}$ represents the stochasticity of the gradient estimate for each $k \in [K]$ and $t \in \{0, 1, \ldots, T-1\}$. The variable $\bar{y}^t$ is designed to estimate the inner function $1/K \sum_{k=1}^K g_k(x)$ in (2). A standard approach to estimate $g_k(x)$ locally for each $k \in [K]$ is to utilize a large batch such that the gradient bias from the inner function estimate can be controlled Guo et al. (2022); Huang et al. (2021); Haddadpour et al. (2022). In contrast, we adopt a momentum-based estimate of $g_k(\cdot)$ at each client $k \in [K]$ that leads to a small bias asymptotically Chen et al. (2021). We note that the estimator utilizes a hybrid estimator that combines a SARAH Nguyen et al. (2017) and SGD Ghadimi & Lan (2013) estimate for the function values rather than the gradients Cutkosky & Orabona (2019). Specifically, individual $y_k^t$'s are estimated in Step 6 as

$$y_k^t = (1 - \beta^t)\Big(y_k^{t-1} - g_k(x_k^{t-1}; \zeta_k^t)\Big) + g_k(x_k^t; \zeta_k^t). \tag{8}$$

for all $k \in [K]$ and where $\beta^t \in (0, 1)$ is the momentum parameter. Motivated by the discussion in Section 4.1, the parameters $y_k^t \in \mathbb{R}^{d_g}$ are shared with the server after the $y_k^t$ update, however, this sharing will not incur a significant communication cost since $y_k^t$'s are usually low dimensional as illustrated in Section 2.1 for DRO problems. The model parameters are then updated using the SG evaluated using (7). Finally, after $I$ local updates the model parameters are aggregated at the server and shared with the clients after aggregation in Step 8. Next, we state the convergence guarantees.

### 4.2.1 MAIN RESULT: CONVERGENCE OF FEDDRO

In the next theorem, we first state the main result of the paper detailing the convergence of FedDRO.

**Theorem 4.3** (Convergence of FedDRO). *For Algorithm 2, choosing the step-size $\eta^t = \eta = \mathcal{O}(\sqrt{K/T})$, the momentum parameter $\beta = 4B_g^4 L_f^2 \eta$ for all $t \in \{0, 1, \ldots, T-1\}$, and $I \leq \mathcal{O}(T^{1/4}/K^{3/4})$. For $T \geq T_{th}$ where $T_{th}$ is defined in Appendix F, then under Assumptions 3.2, 3.3 and 3.4 for $\bar{x}^{a(T)}$ chosen According to Algorithm 2, we have*

$$\mathbb{E}\|\nabla\Phi(\bar{x}^{a(T)})\|^2 \leq \underbrace{\mathcal{O}\Big(\frac{1}{\sqrt{KT}}\Big)}_{\text{Initialization}} + \underbrace{\frac{C_{\sigma_h}\sigma_h^2 + C_{\sigma_g}\sigma_g^2}{\sqrt{KT}}}_{\text{Variance}} + \underbrace{\frac{C_{\Delta_h}\Delta_h^2 + C_{\Delta_g}\Delta_g^2}{\sqrt{KT}}}_{\text{Heterogeneity}},$$

*where $\mathcal{C}(K, T, I) := \max\big\{K(I-1)^2/T, 1/\sqrt{KT}\big\}$ and constants $C_{\sigma_h}$, $C_{\sigma_g}$, $C_{\Delta_h}$, and $C_{\Delta_g}$ are defined in Appendix F.*

We note that the condition on $T \geq T_{th}$ is required for theoretical purposes. Specifically, it ensures that the step-size $\eta = \mathcal{O}(\sqrt{K/T})$ is upper-bounded. A similar requirement has also been posed in Yu et al. (2019a;b); Khanduri et al. (2021) in the past. Theorem 4.3 captures the effect of heterogeneity, stochastic variance, and the initialization on the performance of FedDRO. Theorem 4.3 also states that there exists a choice of the number of local updates that guarantee that FedDRO achieves the same convergence performance as a standard FedAvg Karimireddy et al. (2019); Woodworth et al. (2020); Yu et al. (2019a); Khanduri et al. (2021) for solving the non-CO problems. Next, we characterize the sample and communication complexities of FedDRO.

**Corollary 4.4** (Sample and Communication Complexities). *Under the setting of Theorem 4.3 and choosing the number of local updates as $I = \mathcal{O}(T^{1/4}/K^{3/4})$ the following holds*

*(i) The **sample complexity** of FedDRO is $\mathcal{O}(\epsilon^{-2})$. This implies that each client requires $\mathcal{O}(K^{-1}\epsilon^{-2})$ samples to reach an $\epsilon$-stationary point achieving linear speed-up.*

*(ii) The **communication complexity** of FedDRO is $O(\epsilon^{-3/2})$.*

The sample and communication complexities guaranteed by Corollary 4.4 match that of the standard FedAvg Yu et al. (2019b) for solving stochastic non-convex non-CO problems. We note that in addition to the $O(\epsilon^{-3/2})$ communication complexity that measures the sharing of high-dimensional parameters, FedDRO also shares $\mathcal{O}(K^{-1}\epsilon^{-2})$ low-dimensional embeddings (usually scalar values as illustrated in Section 2.1). Therefore, the total real values shared by each client during the execution of FedDRO is $\mathcal{O}(\epsilon^{-3/2}d + K^{-1}\epsilon^{-2})$. Notice that for high-dimensional models like training (large) neural networks, we will usually have $dK \geq \mathcal{O}(\epsilon^{-0.5})$ meaning the total communication will be $\mathcal{O}(\epsilon^{-3/2})$ which is better than any Federated CO algorithm proposed in the literature Huang et al.

---

**Algorithm 3** Federated non-convex CO algorithm with 2-Sided Learning Rate: DS-FedDRO

1: **Input**: Parameters: $\{\beta^t\}_{t=0}^{T-1}$, $\{\eta^t\}_{t=0}^{T-1}$, $I$, $\gamma_x$, $\gamma_y$
2: **Initialize**: $x_k^{-1} = x_k^0 = x^\tau$, $y_k^0 = y^\tau$ with $\tau = 0$, $\forall k \in [K]$
3: **for** $t = 0$ to $T - 1$ **do**
4:     **for** $k = 1$ to $K$ **do**
5:        `Local Updates:` $\begin{cases} \text{Compute } \nabla\Phi_k(x_k^t; \bar{\xi}_k^t) \text{ using (7)} \\ x_k^{t+1} = x_k^t - \eta^t \nabla\Phi_k(x_k^t; \bar{\xi}_k^t) \\ y_k^{t+1} = (1 - \beta^t)y_k^t + \beta g_k(x_k^{t+1}; \zeta_k^{t+1}) \end{cases}$
6:        **if** $t + 1 \bmod I = 0$ **then**
7:        `Aggregation at Server:` $\begin{cases} x^{\tau+1} = x^\tau - \gamma_x \frac{1}{K}\sum_{k=1}^K (x^\tau - x_k^{t+1}) \\ x_k^{t+1} = x^{\tau+1}, \ \forall k \in [K] \\ y^{\tau+1} = y^\tau - \gamma_y \frac{1}{K}\sum_{k=1}^K (y^\tau - y_k^{t+1}) \\ y_k^{t+1} = y^{\tau+1}, \ \forall k \in [K] \end{cases}$
8:          $\tau = \tau + 1$
9:        **end if**
10:     **end for**
11: **end for**
12: **Return:** $\bar{x}^{a(T)}$ where $a(T) \sim \mathcal{U}\{1, ..., T\}$.

---

(2021); Gao et al. (2022); Guo et al. (2022). Importantly, to our knowledge this is the first work that ensures linear speed up in a federated CO setting, moreover, FedDRO achieves this performance without relying on the computation of large batch sizes. However, to make FedDRO fully federated it is desirable to develop an algorithm that can circumvent the need to communicate the sequences $y_k^t$ at each time instant. Next, we tackle this challenge and develop a novel 2-sided learning rate algorithm DS-FedDRO that avoids the need for frequent communication of $y_k^t$'s.

## 5 DS-FEDDRO: FEDDRO WITH 2-SIDED LEARNING RATE

In this section, we propose a novel algorithm called DS-FedDRO (FedDRO with double-sided learning rates) that relies on the 2-sided learning rate utilized in classical FL algorithms to improve both the experimental and the theoretical performance Yang et al. (2021); Reddi et al. (2020). Importantly, we establish that DS-FedDRO completely avoids the communication of sequence $y_k^{t+1}$ as required by FedDRO while at the same time achieving improved communication complexity. The steps of DS-FedDRO are listed in Algorithm 3. Let us point out a few key differences compared to FedDRO. First, note in Step 8 that instead of performing simple aggregation, the algorithm relies on a 2-sided learning rate update rule for both the $x$- and the $y$-update. Second, note that the 2-sided learning update rule also allows us to update the sequence $y$ utilizing only a single stochastic gradient computation in Step 6. In contrast, FedDRO required two stochastic gradient computations to update $y$. In effect, DS-FedDRO, not only reduces the communication complexity but also improves the per iteration computation complexity over FedDRO. In the following, we present the convergence guarantees of DS-FedDRO and contrast them to that achieved by FedDRO.

### 5.1 MAIN RESULTS: CONVERGENCE OF DS-FEDDRO

For presenting the theoretical results of this section, we utilize a different notion of heterogeneity compared to Assumption 3.4.

**Assumption 5.1** (Bounded Heterogeneity). The heterogeneity of $g_k(\cdot)$ is characterized as $\sup_{x \in \mathbb{R}^d} \|g_k(x) - g(x)\|^2 \leq \Delta_g^2$, for some $\Delta_g > 0$ and for all $k \in [K]$.

Assumption 5.1 above is similar to (Huang et al., 2023, Assumption 5) and (Yang et al., 2024, Assumption 4) for solving bilevel optimization problems with quadratic lower level objective function. Note that this assumption although strong is commonplace in optimization literature and is motivated by the bounded gradient heterogeneity assumptions often made in FL literature (Yu et al., 2019b; Karimireddy et al., 2019; Zhang et al., 2021). Next, we state the main result of the section.

**Theorem 5.2.** *For Algorithm 3, choosing the local step-sizes $\eta^t = \eta = \mathcal{O}(\sqrt{I/T})$ and the momentum parameter $\beta = c_\beta \eta$ for all $t \in \{0, 1, \ldots, T - 1\}$. Choosing the server step-sizes*

$\gamma_x = \mathcal{O}(\sqrt{K/T})$ *and* $\gamma_y = c_{\gamma_y}\gamma_x$. *Then under Assumptions 3.2, 3.3, 3.4, and 5.1 for* $\bar{x}^{a(T)}$ *chosen According to Algorithm 3, we have*

$$\mathbb{E}\big\|\nabla\Phi(\bar{x}^{a(T)})\big\|^2 \le \mathcal{C}_{Sync}\mathcal{O}\left(\sqrt{\frac{1}{KT}}\right) + \mathcal{C}_{Drift}\mathcal{O}\left(\frac{1}{T}\right),$$

*for some constants* $c_\beta$, $c_{\gamma_y}$, $\mathcal{C}_{Sync}$ *and* $\mathcal{C}_{Drift}$.

Again choosing the optimal number of local updates $I$ to minimize the communication complexity of DS-FedDRO. We get the following result.

**Corollary 5.3** (Sample and Communication Complexities). *Under the setting of Theorem 5.2 and choosing the number of local updates as* $I = \mathcal{O}(1/\epsilon)$ *the following holds*

*(i) The **sample complexity** of DS-FedDRO is* $\mathcal{O}(\epsilon^{-2})$. *This implies that each client requires* $\mathcal{O}(K^{-1}\epsilon^{-2})$ *samples to reach an* $\epsilon$-*stationary point achieving linear speed-up.*

*(ii) The **communication complexity** of DS-FedDRO is* $O(\epsilon^{-1})$.

First, note that DS-FedDRO in addition to achieving linear speed-up also improves the communication performance compared to FedDRO. Moreover, it is important to note that the communication complexity of $O(\epsilon^{-1})$ matches the best-known communication complexity even for standard FL problems Zhang et al. (2021); Acar et al. (2020). Moreover, compared to bilevel optimization algorithms the update rules employed by DS-FedDRO (and FedDRO) are much simpler and require the sharing of fewer sequences, thereby, making DS-FedDRO communication efficient compared to such algorithms Tarzanagh et al. (2022); Yang et al. (2024); Li et al. (2024); Huang et al. (2023).

**Comparison of** DS-FedDRO **to** FedDRO. Although DS-FedDRO performs significantly better compared to FedDRO in terms of communication performance, there are some drawbacks of DS-FedDRO that we highlight here. *(i) Additional tuning parameters.* From a practical perspective, because of the addition of server-side learning rates for both $x$- and $y$- updates, DS-FedDRO requires more parameters to tune compared to FedDRO. *(ii) Strong assumptions.* From a theoretical perspective, the improved performance of DS-FedDRO is also made possible with stronger assumptions compared to FedDRO. For example, the analysis of DS-FedDRO relies on additional Assumption 5.1 which FedDRO does not.

## 6 EXPERIMENTS

In this section, we evaluate the performance of FedDRO and DS-FedDRO with both centralized and distributed baselines. Our goal is to 1) establish the superior performance of FedDRO and DS-FedDRO compared to popular federated DRO baselines, and 2) evaluate the performance of FedDRO and DS-FedDRO with different numbers of local updates to capture the effect of data heterogeneity. To evaluate the performance of FedDRO and DS-FedDRO, we focus on two tasks: classification with an imbalanced dataset and learning with fairness constraints. For the first task, we use CIFAR10-ST and CIFAIR-100-ST datasets Qi et al. (2020b) (unbalanced versions of CIFAR10 and CIFAR100 Krizhevsky et al. (2009)) for image classification, and the performance is measured by training and testing accuracy achieved by different algorithms. For the second task, we use the Adult dataset Dua & Graff (2017) for enforcing equality of opportunity (on protected classes) on tabular data classification Hardt et al. (2016). For this setting, the performance is evaluated by training/testing accuracy, and the constraint violations, which are measured by the gap between the true positive rate of the overall data and the protected groups Haddadpour et al. (2022). Please see Appendix B for further details of the classification problem, datasets, experiment settings, and additional experimental evaluation.

**Baseline methods.** For the CIFAR10-ST and CIFAR100-ST datasets we compare FedDRO and DS-FedDRO with popular centralized baselines for classification with imbalanced data. The baselines adopted for comparison are a popular DRO method, FastDRO Levy et al. (2020), a primal-dual SGD approach to solve constrained problems with many constraints, PDSGD Xu (2020), and a popular baseline minibatch SGD, MBSGD, customized for CO Ghadimi & Lan (2013). For the

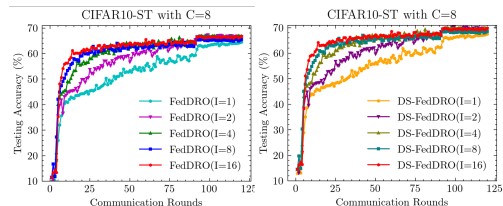

Figure 2: FedDRO and DS-FedDRO on the CIFAR10-ST (100-ST) for different $I$.

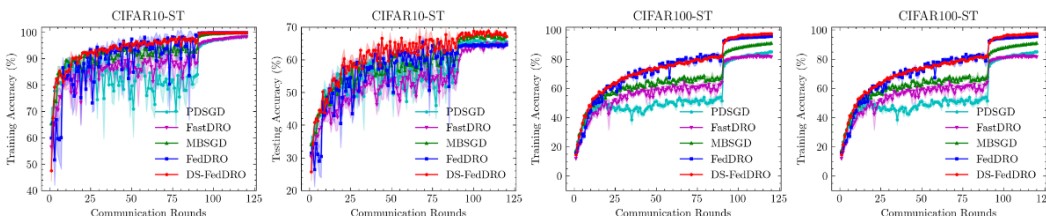

Figure 1: Train and test accuracy vs communication rounds for CIFAR10-ST and CIFAR100-ST.

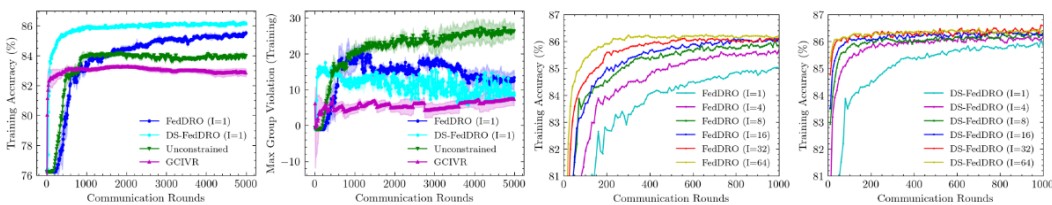

Figure 3: Comparison of FedDRO, DS-FedDRO, GCIVR, and the unconstrained baseline (left two figures), along with the performance of FedDRO and DS-FedDRO across different $I$ values (right two figures).

adult dataset, we use GCIVR Haddadpour et al. (2022) as the baseline distributed model to compare with FedDRO and DS-FedDRO, since like these, it is the only algorithm that can deal with compositional and non-compositional objectives simultaneously. We also implement a parallel SGD as a baseline that ignores the fairness constraints, referred to as unconstrained in the experiments.

**Implementation details.** We use 8 clients to model the distributed setting and split the (unbalanced) dataset equally for each client. We use ResNet20 for classification tasks on CIFAR10-ST and CIFAR100-ST datasets. For a fair comparison with centralized baselines, we choose $I = 1$ for FedDRO and implement a parallel version of the centralized algorithms where the overall gradient computation is $K$ times larger for each algorithm. This is to make sure that the overall gradient computations in each step are uniform across all algorithms. Performance with different values of $I$ is evaluated separately. For each algorithm, we used a batch size of 16 per client, and the learning rates were tuned from the set $\{0.001, 0.01, 0.05, 0.1\}$, the learning rate was dropped to $1/10^{\text{th}}$ after 90 communication rounds. As for the 2-sided learning rates for DS-FedDRO we select 1.3 and 1.4 for the respective tasks. For fairness-constrained classification on the Adult dataset, we use a logistic regression model. For this experiment, we adopt the parameter settings suggested in Haddadpour et al. (2022), for FedDRO and DS-FedDRO we keep the same setting as in the earlier task. All results are averaged over 5 independent runs.

**Discussion.** In Figure 1, we evaluate the performance of FedDRO and DS-FedDRO against the parallel implementations of the centralized baselines on unbalanced CIFAR datasets. Note that FedDRO and DS-FedDRO provide superior training and comparable test accuracy to the state-of-the-art methods, while DS-FedDRO performs even better than FedDRO. In Figure 2, we evaluate the test performance of FedDRO and DS-FedDRO for different number of local updates, $I$. Note that as $I$ increases the performance improves, however, beyond a certain, $I$, the performance doesn't improve capturing the effect of client drift because of data heterogeneity. Finally, in Figure 3 we assess the test performance of FedDRO and DS-FedDRO against the distributed baseline GCIVR on the Adult dataset. We observe that both FedDRO and DS-FedDRO outperform both GCIVR and unconstrained formulation in terms of accuracy and matches the constraint violation performance of GCIVR as communication rounds increase. Finally, for the right two images we evaluate the performance of FedDRO and DS-FedDRO with different values of $I$, we notice that increasing the value of $I$ leads to improved performance, however, beyond a certain threshold (approximately over 32), the performance saturates as a consequence of client drift.

**Conclusion and limitations.** In this work, we first established that vanilla FedAvg algorithms are incapable of solving CO problems in the FL setting. To address this challenge, we showed that either additional communication (FedDRO) or 2-sided learning rate (DS-FedDRO) algorithms are required to guarantee the theoretical convergence of federated CO algorithms. We developed Fed-DRO and DS-FedDRO and performed a thorough theoretical analysis of the two algorithms. We discussed the limitations of each algorithm and established their strong empirical performance via numerical experiments on different tasks.

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

APPENDIX

**Notations.** The expected value of a random variable (r.v) $X$ is denoted by $\mathbb{E}[X]$. Conditioned on an event $\mathcal{F}$ the expectation of a r.v $X$ is denoted by $\mathbb{E}[X|\mathcal{F}]$. We denote by $\mathbb{R}$ (resp. $\mathbb{R}^d$) the real line (resp. the $d$ dimensional Euclidean space). We denote by $[K] := \{1, \dots K\}$. The notation $\| \cdot \|$ defines a standard $\ell_2$-norm. For a set $B$, $|B|$ denotes the cardinality of $B$. We use $\xi \sim \mathcal{D}_h$ and $\zeta \sim \mathcal{D}_g$ to denote the stochastic samples of functions $h(\cdot)$ and $g(\cdot)$ from distributions $\mathcal{D}_h$ and $\mathcal{D}_g$, respectively. A batch of samples from $h(\cdot)$ (resp. $g(\cdot)$) is denoted by $b_h$ (resp. $b_g$). Moreover, joint samples of $h(\cdot)$ and $g(\cdot)$ are denoted by $\bar{\xi} = \{b_h, b_g\}$. We represent by $\bar{x}$ the empirical average of a sequence of vectors $\{x_k\}_{k=1}^K$.

## A  RELATED WORK

**Centralized CO.** The first non-asymptotic analysis of stochastic CO problems was performed in Wang et al. (2017) where the authors proposed SCGD a two-timescale algorithm for solving the problem (1). The convergence of SCGD was improved in Wang et al. (2016) where the authors proposed an accelerated variant of SCGD. Both SCGD and its accelerated variant achieved convergence rates that were strictly worse than those of SGD for solving non-CO problems. Recently, Ghadimi et al. (2020) and Chen et al. (2021) developed a single time-scale algorithm for solving the CO problem that achieves the same convergence as SGD for solving non-CO problems. Variance-reduced algorithms for solving the CO problems have also been considered in the literature, however, a major drawback of such approaches is the reliance of batch size on the desired solution accuracy Lian et al. (2017); Zhang & Xiao (2019); Hu et al. (2019).

**Distributed CO.** There have been only a few attempts to solve non-convex CO problems in the FL setting, partially, because of the challenges discussed in Section 1. The first FL algorithm to solve the non-convex CO problem, Compositional Federated Learning (ComFedL), was developed in Huang et al. (2021). ComFedL required accuracy-dependent batch sizes that resulted in $\mathcal{O}(\epsilon^{-4})$ convergence which is significantly worse compared to FedAvg to solve standard non-compositional problems Yu et al. (2019b). In Gao et al. (2022), Local Stochastic Compositional Gradient Descent with Momentum (Local-SCGDM) was proposed which removed the requirement of large batch sizes and achieved an $\mathcal{O}(\epsilon^{-2})$ convergence. However, Local-SCGDM utilized a non-standard momentum-based update from Ghadimi et al. (2020) that does not resemble a simple SGD-based update. Importantly, the CO problem solved by ComFedL Huang et al. (2021) and Local-SCGDM Gao et al. (2022) is non-standard as the problem is not distributed in the compositional objective (see Remark 2.1). In contrast, we consider a general setting where the compositional objective is also distributed among multiple nodes. Recently, Tarzanagh et al. (2022) proposed a nested optimization framework, FedNest, to solve bilevel problems in the FL setting. The proposed algorithm achieved SGD rates of $\mathcal{O}(\epsilon^{-2})$ Ghadimi & Lan (2013). Different from the simple SGD-based update rule, FedNest adopted a multi-loop variance reduction-based update. In Haddadpour et al. (2022), the authors proposed a Generalized Composite Incremental Variance Reduction (GCIVR) framework for solving problems of the form (2) in a distributed setting. GICVR achieved a better convergence rate of $\mathcal{O}(\epsilon^{-1.5})$, however, it relied on a double-loop structure and accuracy-dependent large batch sizes to achieve variance reduction. Importantly, none of the above works guarantee linear speedup with the number of clients. Moreover, the current algorithms utilize complicated momentum or VR-based update rules that require computation of accuracy-dependent batch sizes Haddadpour et al. (2022), and/or consider a simple setting where the compositional objective is not distributed among nodes Huang et al. (2021); Gao et al. (2022).

In contrast to all the above works, our work considers a general setting (2), where the goal is to jointly minimize a compositional and a non-compositional objective in the FL setting. To solve (2), we develop FedDRO a FedAvg algorithm for CO problems that achieves (i). the same guarantees as FedAvg for minimizing non-CO problems, (ii). linear speed-up with the number of clients, (iii). improved communication complexity, (iv). performance guarantees where the batch sizes required are independent of the desired solution accuracy, and (v). characterizes the performance as a function of local updates at each client and the data heterogeneity in the inner and outer non-compositional objectives.

**DRO.** DRO has been extensively studied in optimization, machine learning, and statistics literature Ben-Tal et al. (2013); Bertsimas et al. (2018); Duchi et al. (2021); Namkoong & Duchi (2017); Staib & Jegelka (2019) Broadly, DRO problem formulation can be divided into two classes, one is a constrained formulation and the other is the regularized formulation (see (3)) Levy et al. (2020); Duchi et al. (2021). A popular approach to solve the constrained DRO formulation is via primal-dual formulation where algorithms developed for min-max problems can directly be applied to solve constrained DRO Yan et al. (2019); Namkoong & Duchi (2017); Song et al. (2021); Alacaoglu et al. (2022); Tran Dinh et al. (2020). Many algorithms under different settings, e.g., convex, non-convex losses, and stochastic settings have been considered in the past to address such problems. However, primal-dual algorithms suffer from computational bottlenecks, since they require maintaining and updating the set of dual variables equal to the size of the dataset which can become particularly challenging, especially for large-scale machine learning tasks. Recently, Levy et al. (2020) Qi et al. (2022) Haddadpour et al. (2022) have developed algorithms that are applicable to large-scale stochastic settings. Works Levy et al. (2020) and Qi et al. (2022) consider specific formulations of the DRO problem while Haddadpour et al. (2022) considers a general formulation, however, as pointed out earlier the algorithms developed in Haddadpour et al. (2022) are double loop and require accuracy-dependent batch sizes to guarantee convergence (see Table 1). In contrast, in this work, we develop algorithms that solve general instants of CO problems that often arise in DRO formulation. Importantly, the developed algorithms are amenable to large-scale distributed implementation with algorithmic guarantees independent of accuracy-dependent batch sizes.

## A.1 Detailed Comparison with Huang et al. (2021); Gao et al. (2022); Tarzanagh et al. (2022)

**Comparison with Huang et al. (2021); Gao et al. (2022).** We note that the problem setting in Huang et al. (2021) and Gao et al. (2022) is significantly different from the one considered in our work. We also would like to point out that the problem formulation considered in our work is more challenging than Huang et al. (2021); Gao et al. (2022) and the algorithms developed for solving the problem in Huang et al. (2021); Gao et al. (2022) cannot solve the problem considered in our work. In the following, we elaborate on the differences between our work and that of Huang et al. (2021); Gao et al. (2022).

In Huang et al. (2021); Gao et al. (2022), the authors consider the objective function

$$\frac{1}{K} \sum_{k=1}^{K} f_k(g_k(\cdot)). \tag{9}$$

Please observe that in this setting the local nodes have access to local composite functions $f_k(g_k(\cdot))$. In contrast, we consider a setting with objective function defined in (2) where the local nodes have access to only $h_k(\cdot)$ and $g_k(\cdot)$[1]. Note that the major difference in the two settings in (9) and (2) comes from the fact that in (9) the inner function $g_k(\cdot)$ is fully available at each node, whereas in (2) the inner function $1/K \sum_{k=1}^{K} g_k(\cdot)$ is not available (since each node can only access $g_k(\cdot)$) at the local nodes. Below, we discuss two major consequences of this:

- **Practicality:** We point out that the setting in (2) is more practical as can be seen from the examples presented in Section 2.1 wherein the DRO problems take the form of (2) rather than (9) in a distributed setting. For illustration, let us consider a simple setting where we have a total of $m$ samples with each node having access to $m_k = m/K$ samples. Then the DRO problem with KL-Divergence problem becomes

$$\min_{x \in \mathbb{R}^d} f\left(\frac{1}{K} \sum_{k=1}^{K} g_k(\cdot)\right) := \log\left(\frac{1}{m} \sum_{i=1}^{m} \exp\left(\frac{\ell_i(x)}{\lambda}\right)\right),$$

where $f(\cdot) = \log(\cdot)$, $g_k(x) = 1/m_k \sum_{i=1}^{m_k} \exp\left(\ell_i(x)/\lambda\right)$, and $g(\cdot) = 1/K \sum_{k=1}^{K} g_k(\cdot)$. Note that the above formulation is same as (2) and cannot be formulated using (9). To demonstrate this fact we have used the notation in Table 1 as CO-ND for formulation of (9 where the inner

---

[1]We would also like to note that the setting considered in the paper can be easily extended to the case where $f(\cdot) = 1/K \sum_{k=1}^{K} f_k(\cdot)$ without changing the current results.

function $g_k(\cdot)$ can be fully locally accessed by each node whereas our setting is more general with each node having only partial access to the inner-function $g(\cdot)$. Next, we show why the algorithms developed for Huang et al. (2021); Gao et al. (2022) cannot be utilized to solve the problem considered in our work.

- **Challenges in solving (2):** A major contribution of our work is in establishing the fact that the algorithms that are developed for solving 2), i.e., the algorithms developed in Huang et al. (2021); Gao et al. (2022), cannot be utilized to solve the problem considered in our work.

  To demonstrate this consider the simple deterministic setting with $f_k = f$, then the local gradient computed for the objective function in (2) will be $\nabla g_k(x)\nabla f(g_k(x))$ (please see (6) in the manuscript). Note that this is an unbiased local gradient for objective in (9) which further implies that simple FedAVG-based implementations can be developed for solving this problem as done in Huang et al. (2021); Gao et al. (2022). In contrast, note that the local gradient $\nabla g_k(x)\nabla f(g_k(x))$ will be a biased local gradient for our problem in (2) and will lead to divergence of FedAvg-based algorithms Huang et al. (2021); Gao et al. (2022) as shown in Section 4.1. Moreover, note that we establish that even if we share the local functions $g_k(\cdot)$ intermitteltly among nodes we may not be able to mitigate the bias of local gradient and the developed algorithms will again diverge to incorrect solutions. Please see Section 4.1 for more details.

**Comparison with Tarzanagh et al. (2022).** Next, we note that the algorithm deveoped in Tarzanagh et al. (2022) is a bilevel algorithm with multi-loop structure with many tunable (hyper) parameters. Such algorithms are not preferred in practical implementations. In contrast our algorithm is a single-loop algorithm with simple FedAvg-type SGD updates. In addition to being practical, our work also significantly improves upon the theoretical guarantees achieved in Tarzanagh et al. (2022) by achieving linear speed-up with the number of clients as well as improved communication complexity which any of the works including Huang et al. (2021); Gao et al. (2022); Tarzanagh et al. (2022) are unable to achieve.

# B   DETAILED EXPERIMENT SETUP AND ADDITIONAL EXPERIMENTS

**Experiment setup.** The models are trained on an NVIDIA GeForce RTX 3090 GPU with 24 GB of memory. All experiments are conducted using the PyTorch framework, specifically Python 3.9.16 and PyTorch 1.8

**Datasets.** To evaluate the performance of FedDRO and DS-FedDRO, the first section of the experiments is conducted on CIFAR10-ST and CIFAR-100-ST datasets for image classification. The second section of the experiments focuses on the Adult dataset, utilizing tabular data classification and emphasizing DRO for fairness constraints. The CIFAR10-ST and CIFAR-100-ST datasets are modified versions of the original CIFAR10 and CIFAR-100 datasets. The modification involves intentionally creating imbalanced training data. Specifically, only the last 100 images are retained for each class in the first half of the classes, while the other classes and the test data remain unchanged. This creates an imbalanced distribution, posing a challenge for machine learning models to effectively handle imbalanced class scenarios. In the Adult dataset, we consider the race groups "white," "black," and "other" as protected groups. We assign the value of $\epsilon$ as 0.05 and set the noise level to 0.3 during training across all the algorithms.

**Evaluation metrics.** We present the Top-1 accuracies for the training and testing segments of the CIFAR10-ST and CIFAR-100-ST datasets (please see Figures 1 and 2 in Section 6 and Figure 4 in Section B). Furthermore, in addition to training and testing performance, we also include the maximum violation values for both the training and testing sections of the Adult dataset. Specifically, the maximum group violation is evaluated following Haddadpour et al. (2022). To ensure equal opportunities among different groups, even when group membership is uncertain and fluctuating during training, the objective is to develop a solution that is robust across various protected

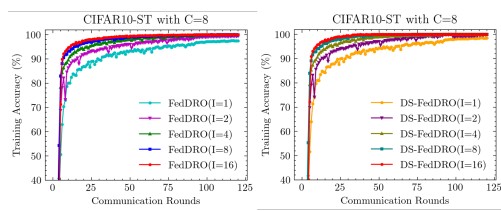

Figure 4: Training accuracy of FedDRO and DS-FedDRO on the CIFAR10-ST and CIFAR100-ST for different $I$.

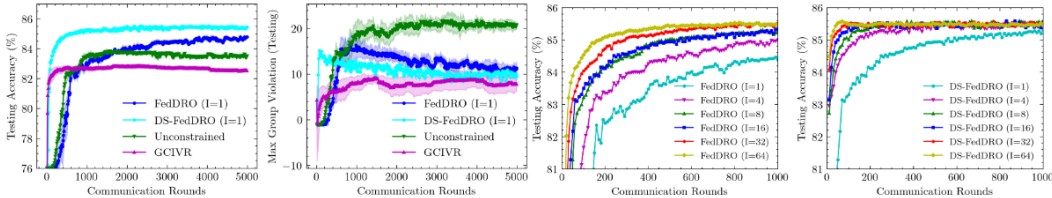

Figure 5: Overall training performance comparison of FedDRO, DS-FedDRO, GCIVR, and the unconstrained baseline (left two figures), along with the performance of FedDRO and DS-FedDRO across different $I$ values (right two figures).

groups in the problem. We assume that we have access to the probability distribution of the actual group memberships ($P(gi = j|g^i = k)$ where $g^i$ represents the true group membership and $g^i$ represents the noisy group membership). With this information, we aim to enforce fairness constraints by considering all potential proxy groups based on this probability distribution, which can significantly increase the number of constraints. In the case of equal opportunity, our goal is to ensure that the true positive rate ($TPR$) for each group closely aligns with the $TPR$ of the overall dataset, within a certain threshold $\epsilon$. In other words, we want to achieve $tpr(g = j) \geq tpr(ALL) - \epsilon$ for every proxy group we define.

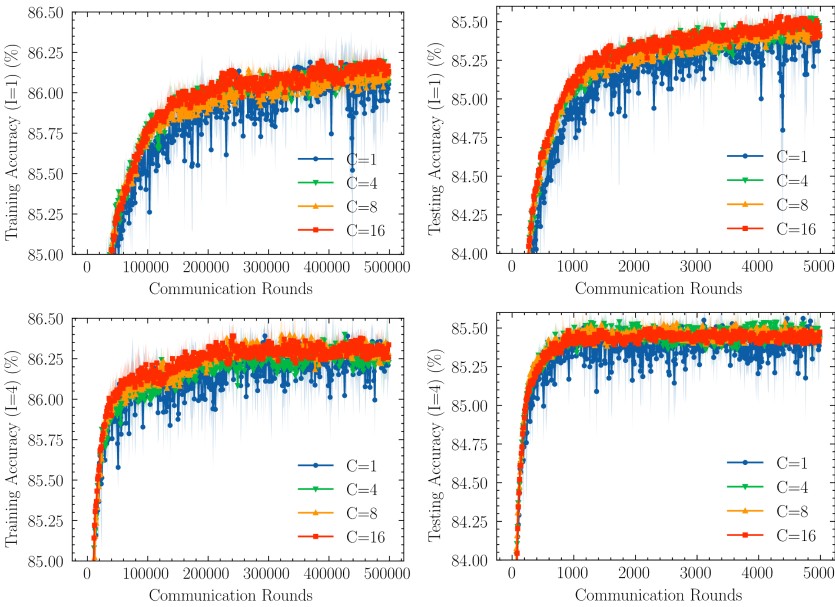

Figure 6: Training and testing performance of FedDRO with the number of clients (denoted as $C = 1, 2, 3$ and $4$ in the figure) and number of local updates, $I = 1$ and $4$.

**Discussion.** In Figure 4, we assess the performance of FedDRO and DS-FedDRO on the training dataset under the same conditions described in Section 6, but with varying numbers of local updates, $I$. It is observed that as $I$ increases, performance improves; however, beyond a certain point, further increases in $I$ do not lead to improvement, highlighting the impact of client drift due to data heterogeneity. In Figure 5, we evaluate the training performance on the adult dataset under the same conditions as mentioned earlier for testing in Section 6. Similar to the previous findings, in the leftmost image, we observe that FedDRO and DS-FedDRO outperform both the constrained version of GCIVR and unconstrained baseline formulation while FedDRO outperforming DS-FedDRO easily. Evaluating the maximum group violation, we see that the unconstrained optimization demonstrates the poorest performance, while our techniques perform comparably to GCIVR and improve performance as the communication rounds increase. The right two plots, confirm that increasing

the local updates, i.e., $I$ results in improved performance, aligning with the theoretical guarantees presented in the paper.

In Figure 6, we evaluate the performance of FedDRO with the number of clients. Specifically, the accuracy demonstrates an upward trend as the value of $C$ (representing the number of clients) increases in the experiments conducted on the adult dataset. The top two plots depict the training and testing performance for $I = 1$, while the bottom two demonstrate the training and testing performance with $I = 4$.

## C    USEFUL LEMMAS

**Lemma C.1.** *For vectors $a_1, a_2, \ldots, a_n \in \mathbb{R}^d$, we have*

$$\|a_1 + a_2 + \ldots, +a_n\|^2 \le n\big[\|a_1\|^2 + \|a_2\|^2 + \ldots, +\|a_n\|^2\big].$$

**Lemma C.2.** *For a sequence of vectors $a_1, a_2, \ldots, a_K \in \mathbb{R}^d$, defining $\bar{a} := \frac{1}{K}\sum_{k=1}^K a_k$, we then have*

$$\sum_{k=1}^K \|a_k - \bar{a}\|^2 \le \sum_{k=1}^K \|a_k\|^2.$$

## D    PROOF OF THEOREM 4.1

We restate Theorem 4.1 for convenience.

**Theorem D.1** (Vanilla FedAvg: Non-Convergence for CO). *There exist functions $f(\cdot)$ and $g_k(\cdot)$ for $k \in [K]$ satisfying Assumptions 3.2, 3.3, and 3.4, and an initialization strategy such that for a fixed number of local updates $I > 1$, and for any $0 < \eta^t < C_\eta$ for $t \in \{0, 1, \ldots, T-1\}$ where $C_\eta > 0$ is a constant, the iterates generated by Algorithm 1 under both Cases I and II do not converge to the stationary point of $\Phi(\cdot)$, where $\Phi(\cdot)$ is defined in (2) with $h(x) = 0$.*

*Proof.* We consider a setting where we have $K = 2$ nodes in the network. Also, let us consider a single-dimensional setting where the local functions $g_k : \mathbb{R} \to \mathbb{R}$ for $k = \{1, 2\}$ at each node are

$$g_1(x) := 4x - 4 \quad \text{and} \quad g_2(x) := -2x + 4.$$

Moreover, assume $f : \mathbb{R} \to \mathbb{R}$ as $f(y) := \sqrt{y^2 + 4}$. Therefore, the CO problem becomes

$$\min_{x \in \mathbb{R}}\left\{\Phi(x) := f\left(\frac{1}{2}\big(g_1(x) + g_2(x)\big)\right) := \sqrt{\left[\frac{1}{2}\big(g_1(x) + g_2(x)\big)\right]^2 + 4} = \sqrt{x^2 + 4}\right\}. \quad (10)$$

First, we establish that the functions $f(\cdot)$ and $g_k(\cdot)$ for $k \in [K]$ satisfy Assumptions 3.2, 3.3, and 3.4.

**Claim:** Functions $f$, $g_1$ and $g_2$ satisfy Assumptions 3.2, 3.3, and 3.4.

The above claim is straightforward to verify. Specifically, we have

– The functions $f$, $g_1$ and $g_2$ are differentiable and Lipschitz smooth.

– The function $f(\cdot)$ is Lipschitz. Moreover, $g_k(\cdot)$'s are deterministic functions implying mean-squared Lipschitzness.

– Assumption 3.3 is automatically satisfied since $g_k(\cdot)$'s are deterministic functions.

– Bounded heterogeneity of $g_k(\cdot)$'s is satisfied.

Note that it is clear from (10) that the minimizer of $\Phi(\cdot)$ is $x^* = 0$. In the following, we will show that Algorithm 1 is not suitable to solve such problems by establishing that there exists an initialization strategy and choice of step-sizes in the range $0 < \eta < C_\eta$ where $C_\eta > 0$ is a constant,

the iterates generated by Algorithm 1 under both Cases I and II fail to converge to $x^*$. Next, we prove the statement of the theorem in two parts. In the first part, we tackle Case I of Algorithm 1 while in the second part, we prove Case II of Algorithm 1. Next, we consider Case I.

**Case I:** Let us first compute the local gradients at each agent. We have

$$\nabla\Phi_1(x) = \nabla g_1(x)\nabla f(y_1) = 4\frac{y_1}{\sqrt{y_1^2 + 4}}$$

$$\nabla\Phi_2(x) = \nabla g_2(x)\nabla f(y_2) = -2\frac{y_2}{\sqrt{y_2^2 + 4}}$$

To prove the results, we consider a simple setting with $I = 2$, i.e., each node conducts 2 local updates and shares the model parameters with the server. Moreover, we initialize the local iterates to be $x_k^0 = \bar{x}^0 = 0.5$ for $k = \{1, 2\}$ at both nodes. For this setting, let us write the update rule for Algorithm 1 in Case I.

1. Note that for every $t$ such that $t \mod 2 = 0$, the local update at each node will be:

$$x_1^{t+1} = \bar{x}^t - 4\eta\frac{4\bar{x}^t - 4}{\sqrt{(4\bar{x}^t - 4)^2 + 4}}$$

$$x_2^{t+1} = \bar{x}^t + 2\eta\frac{-2\bar{x}^t + 4}{\sqrt{(-2\bar{x}^t + 4)^2 + 4}},$$

2. Moreover, the next immediate update at each node will be

$$x_1^{t+2} = x_1^{t+1} - 4\eta\frac{4x_1^{t+1} - 4}{\sqrt{(4x_1^{t+1} - 4)^2 + 4}}$$

$$x_2^{t+2} = x_2^{t+1} + 2\eta\frac{-2x_2^{t+1} + 4}{\sqrt{(-2x_2^{t+1} + 4)^2 + 4}},$$

3. This process keeps repeating for $T$ iterations.

Let us focus on the local functions $f(g_1(x))$ and $f(g_2(x))$. Note from the definition of $g_1(\cdot)$, $g_2(\cdot)$ and $f(\cdot)$ that the local optimum of these functions will be $x_1^* = 1$ and $x_2^* = 2$, respectively. Consequently, for appropriately chosen step-size $\eta$ in each iteration $x_1^{t+1}$ and $x_1^{t+2}$ at node 1 will converge towards $x_1^* = 1$ and similarly, $x_2^{t+1}$ and $x_2^{t+2}$ at node 2 will converge towards $x_1^* = 2$. This implies that we can expect the sequence $\bar{x}^t$ for each $t \in [T]$ to not converge to $x^* = 0$, the minimizer of the CO problem defined in (10). Let us present this argument formally.

**Claim:** For $C_\eta = 1/8$ such that we have $0 < \eta < C_\eta$, and utilizing the initialization $\bar{x}^0 = 0.5$, we have $\bar{x}^t \geq 0.5$ for every $t > 0$ with $t \mod 2 = 0$.

This above Claim directly proves the statement of Theorem 4.1 for Case I. Let us now prove the claim formally. We utilize induction to prove the claim.

*Proof of claim:* First, note that the claim is automatically satisfied for $t = 0$ as a consequence of the initialization strategy. Assuming the claim holds for some $t \in [T]$ with $t \mod 2 = 0$, i.e., we have $\bar{x}_t \geq 0.5$ for some $t \in [T]$ with $t \mod 2 = 0$, we need to show that $\bar{x}_{t+2} \geq 0.5$.

In the following, we consider the following three cases: (1) $0.5 \leq \bar{x}_t < 1$, (2) $1 \leq \bar{x}_t < 2$, and (3) $\bar{x}_t \geq 2$. Here, we present the proof for case (1), the rest of the cases follow in a similar manner.

- Note from Step 1 above that since $0.5 \leq \bar{x}^t < 1$, we have $4\bar{x}^t - 4 < 0$ and $-2\bar{x}^t + 4 > 0$, which further implies that the locally updated iterates $x_1^{t+1} > \bar{x}^t \geq 0.5$ and $x_2^{t+1} > \bar{x}^t \geq 0.5$. Next, let us analyze the iterates at $t + 2$.

- At node 1, we further consider two cases, when $x_1^{t+1} < 1$ and the other when $x_1^{t+1} \geq 1$.

- First, note that if $x_1^{t+1} < 1$ we will have $4x_1^{t+1} - 4 < 0$ in Step 2 above implying $x_1^{t+2} > x_1^{t+1} > \bar{x}^t \geq 0.5$.

- Otherwise, if $x_1^{t+1} \geq 1$, we have $4x_1^{t+1} - 4 \geq 0$ however in this case we have

$$\left| 4\eta \frac{4x_1^{t+1} - 4}{\sqrt{(4x_1^{t+1} - 4)^2 + 4}} \right| \leq 1/2 \text{ for } \eta \leq \frac{1}{8},$$

again implying from the update rule in Step 2 that

$$x_1^{t+2} \geq x_1^{t+1} - \frac{1}{2} \geq 0.5,$$

where the last step follows from the fact that $x_1^{t+1} \geq 1$. Therefore, we have established that $x_1^{t+2} \geq 0.5$.

- At node 2, it is easy to establish that for case (1) with $0.5 \leq \bar{x}_t < 1$, we will have $0.5 \leq x_2^{t+1} \leq 1.5$. Note from the update rule in Step 2 that for this $x_2^{t+1}$, we have $-2x_2^{t+1} + 4 > 0$ which further implies that $x_2^{t+2} > x_2^{t+1} \geq 0.5$.

- Finally, we have established that both $x_1^{t+2} \geq 0.5$ and $x_2^{t+2} \geq 0.5$, implying $\bar{x}_{t+2} \geq 0.5$. This completes the proof of Case (1). Note that the proof for the other cases follows in a very similar straightforward manner.

Therefore, we have the proof of Case I in Algorithm 1. Next, we consider Case II where in addition to the model parameters, the local embeddings $g_k(\cdot)$ for $k \in [K]$ are also shared intermittently among nodes. Please see Case II in Algorithm 1.

**Case II:** Let us consider the same setting as in Case I. Specifically, we consider a simple setting with $I = 2$, i.e., each node conducts 2 local updates and shares the model parameters with the server. Moreover, we initialize the model parameters $x_k^0 = \bar{x}^0 = 0.5$ for $k = \{1, 2\}$ at both nodes. Note that this implies from the definition of $g_1(\cdot)$ and $g_2(\cdot)$ that $y_k^0 = \bar{y}^0 = 0.5$ for $k = \{1, 2\}$. For this setting, let us write the update rule for Algorithm 1.

1. Note that for every $t$ such that $t \mod 2 = 0$, the local update at each node will be:

$$x_1^{t+1} = \bar{x}^t - 4\eta \frac{\bar{x}^t}{\sqrt{(\bar{x}^t)^2 + 4}}$$

$$x_2^{t+1} = \bar{x}^t + 2\eta \frac{\bar{x}^t}{\sqrt{(\bar{x}^t)^2 + 4}},$$

2. Moreover, the next immediate update at each node will be

$$x_1^{t+2} = x_1^{t+1} - 4\eta \frac{4x_1^{t+1} - 4}{\sqrt{(4x_1^{t+1} - 4)^2 + 4}}$$

$$x_2^{t+2} = x_2^{t+1} + 2\eta \frac{-2x_2^{t+1} + 4}{\sqrt{(-2x_2^{t+1} + 4)^2 + 4}},$$

3. This process keeps repeating for $T$ iterations.

We point out that this setting is considerably challenging compared to Case I since a cursory look at the algorithm may suggest that sharing the embeddings $g_k(\cdot)$ for $k \in [K]$ intermittently may help mitigate the bias in the gradient estimates. However, this is not the case as we show next.

**Claim:** For $C_\eta = 1/22$ such that we have $0 < \eta < C_\eta$, and utilizing the initialization $\bar{x}^0 = 0.5$, we have $\bar{x}^t \geq 0.5$ for every $t > 0$ with $t \mod 2 = 0$.

We note that for this case the intuition is not as straightforward as in the previous case. We again prove the claim by induction.

*Proof of claim:* First, note that the claim is automatically satisfied for $t = 0$ as a consequence of the initialization strategy. Assuming the claim holds for some $t \in [T]$ with $t \bmod 2 = 0$, i.e., we have $\bar{x}_t \geq 0.5$ for some $t \in [T]$ with $t \bmod 2 = 0$, we need to show that $\bar{x}_{t+2} \geq 0.5$.

Let us first construct $x_1^{t+2}$ and $x_2^{t+2}$ as a function of $\bar{x}^t$. To this end, we have from the update rule in Steps 1 and 2 that

$$x_1^{t+2} = \bar{x}^t\left(1 - \epsilon_1^t\right) - 4\eta \frac{4\bar{x}^t\left(1 - \epsilon_1^t\right) - 4}{\sqrt{\left(4\bar{x}^t\left(1 - \epsilon_1^t\right) - 4\right)^2 + 4}}$$

$$x_2^{t+2} = \bar{x}^t\left(1 + \epsilon_2^t\right) + 2\eta \frac{-2\bar{x}^t\left(1 + \epsilon_2^t\right) + 4}{\sqrt{\left(-2\bar{x}^t\left(1 + \epsilon_2^t\right) + 4\right)^2 + 4}},$$

where we have defined $\epsilon_1^t := \frac{4\eta}{\sqrt{(\bar{x}^t)^2 + 4}}$ and $\epsilon_2^t := \frac{2\eta}{\sqrt{(\bar{x}^t)^2 + 4}}$, therefore, we have $\epsilon_1^t = 2\epsilon_2^t$. Using the above we can evaluate $\bar{x}^{t+2}$ as

$$\bar{x}^{t+2} = \frac{1}{2}\left(x_1^{t+2} + x_2^{t+2}\right)$$

$$= \left(\frac{2 - \epsilon_1^t + \epsilon_2^t}{2}\right)\bar{x}^t + 2\eta \frac{4 - 4\bar{x}^t\left(1 - \epsilon_1^t\right)}{\sqrt{\left(4\bar{x}^t\left(1 - \epsilon_1^t\right) - 4\right)^2 + 4}} + \eta \frac{4 - 2\bar{x}^t\left(1 + \epsilon_2^t\right)}{\sqrt{\left(-2\bar{x}^t\left(1 + \epsilon_2^t\right) + 4\right)^2 + 4}}$$

$$= \left(1 - \frac{\epsilon_2^t}{2}\right)\bar{x}^t + 2\eta \frac{4 - 4\bar{x}^t\left(1 - \epsilon_1^t\right)}{\sqrt{\left(4\bar{x}^t\left(1 - \epsilon_1^t\right) - 4\right)^2 + 4}} + \eta \frac{4 - 2\bar{x}^t\left(1 + \epsilon_2^t\right)}{\sqrt{\left(-2\bar{x}^t\left(1 + \epsilon_2^t\right) + 4\right)^2 + 4}},$$

where in the first term of the last equality, we have used the fact that $\epsilon_1^t = 2\epsilon_2^t$. Recall from the induction hypothesis that we have $\bar{x}^t \geq 0.5$, and we need to show that $\bar{x}^{t+2} \geq 0.5$. Note from above that to establish $\bar{x}^{t+2} \geq 0.5$, it suffices to show that

$$\bar{x}^t - 0.5 + 2\eta \frac{4 - 4\bar{x}^t\left(1 - \epsilon_1^t\right)}{\sqrt{\left(4\bar{x}^t\left(1 - \epsilon_1^t\right) - 4\right)^2 + 4}} + \eta \frac{4 - 2\bar{x}^t\left(1 + \epsilon_2^t\right)}{\sqrt{\left(-2\bar{x}^t\left(1 + \epsilon_2^t\right) + 4\right)^2 + 4}} \geq \frac{\epsilon_2^t}{2}\bar{x}^t. \qquad (11)$$

From the definition of $\epsilon_2^t := \frac{2\eta}{\sqrt{(\bar{x}^t)^2 + 4}}$, we note that the r.h.s. term can be further upper bounded as

$$\frac{\epsilon_2^t}{2}\bar{x}^t = \eta \frac{\bar{x}^t}{\sqrt{(\bar{x}^t)^2 + 4}} \leq \eta.$$

Therefore, to establish to establish $\bar{x}^{t+2} \geq 0.5$, it suffices to show that

$$\bar{x}^t - 0.5 + 2\eta \frac{4 - 4\bar{x}^t\left(1 - 2\epsilon_2^t\right)}{\sqrt{\left(4\bar{x}^t\left(1 - 2\epsilon_2^t\right) - 4\right)^2 + 4}} + \eta \frac{4 - 2\bar{x}^t\left(1 + \epsilon_2^t\right)}{\sqrt{\left(-2\bar{x}^t\left(1 + \epsilon_2^t\right) + 4\right)^2 + 4}} \geq \eta, \qquad (12)$$

where we have replaced $\epsilon_1^t = 2\epsilon_2^t$. Similar to the previous proof here we again consider three cases as listed below

- Case (1): $\frac{4 - 4\bar{x}^t\left(1 - 2\epsilon_2^t\right)}{\sqrt{\left(4\bar{x}^t\left(1 - 2\epsilon_2^t\right) - 4\right)^2 + 4}} < 0$ and $\frac{4 - 2\bar{x}^t\left(1 + \epsilon_2^t\right)}{\sqrt{\left(-2\bar{x}^t\left(1 + \epsilon_2^t\right) + 4\right)^2 + 4}} < 0$

- Case (2): $\frac{4 - 4\bar{x}^t\left(1 - 2\epsilon_2^t\right)}{\sqrt{\left(4\bar{x}^t\left(1 - 2\epsilon_2^t\right) - 4\right)^2 + 4}} < 0$ and $\frac{4 - 2\bar{x}^t\left(1 + \epsilon_2^t\right)}{\sqrt{\left(-2\bar{x}^t\left(1 + \epsilon_2^t\right) + 4\right)^2 + 4}} > 0$

- Case (3): $\frac{4 - 4\bar{x}^t\left(1 - 2\epsilon_2^t\right)}{\sqrt{\left(4\bar{x}^t\left(1 - 2\epsilon_2^t\right) - 4\right)^2 + 4}} \geq 0$ and $\frac{4 - 2\bar{x}^t\left(1 + \epsilon_2^t\right)}{\sqrt{\left(-2\bar{x}^t\left(1 + \epsilon_2^t\right) + 4\right)^2 + 4}} \geq 0$

We first consider Case (1). Note that Case (1) implies that $\bar{x}^t > 1$, and using the fact that $\frac{4-4\bar{x}^t(1-2\epsilon_2^t)}{\sqrt{(4\bar{x}^t(1-2\epsilon_2^t)-4)^2+4}} \geq -1$ and $\frac{4-2\bar{x}^t(1+\epsilon_2^t)}{\sqrt{(-2\bar{x}^t(1+\epsilon_2^t)+4)^2+4}} \geq -1$, we get

$$\bar{x}^t - 0.5 + 2\eta\frac{4-4\bar{x}^t(1-2\epsilon_2^t)}{\sqrt{(4\bar{x}^t(1-2\epsilon_2^t)-4)^2+4}} + \eta\frac{4-2\bar{x}^t(1+\epsilon_2^t)}{\sqrt{(-2\bar{x}^t(1+\epsilon_2^t)+4)^2+4}} \geq 0.5 - 3\eta$$

Note that by choosing $\eta \leq 1/8$, the sufficient condition in (12) is satisfied, which further implies that under Case (1), we have $\bar{x}^{t+2} \geq 0.5$. Next, we consider Case (2).

Note that for Case (2) we have $2/(1+\epsilon_2^t) > \bar{x}^t > 1$, next using the fact that $\frac{4-4\bar{x}^t(1-2\epsilon_2^t)}{\sqrt{(4\bar{x}^t(1-2\epsilon_2^t)-4)^2+4}} \geq -1$ and $\frac{4-2\bar{x}^t(1+\epsilon_2^t)}{\sqrt{(-2\bar{x}^t(1+\epsilon_2^t)+4)^2+4}} \geq 0$, we get

$$\bar{x}^t - 0.5 + 2\eta\frac{4-4\bar{x}^t(1-2\epsilon_2^t)}{\sqrt{(4\bar{x}^t(1-2\epsilon_2^t)-4)^2+4}} + \eta\frac{4-2\bar{x}^t(1+\epsilon_2^t)}{\sqrt{(-2\bar{x}^t(1+\epsilon_2^t)+4)^2+4}} \geq 0.5 - 2\eta$$

Again choosing $\eta \leq 1/8$, the sufficient condition in (12) is satisfied, which further implies that under Case (2), we have $\bar{x}^{t+2} \geq 0.5$.

Finally, we consider the most challenging Case (3). Note that in Case (3) we have $0.5 \leq \bar{x}^t \leq 1/(1-2\epsilon_2^t)$. For this case, we revisit the sufficient condition in (11) and make it tight. Recall that we had from (11) that

$$\bar{x}^t - 0.5 + 2\eta\frac{4-4\bar{x}^t(1-2\epsilon_2^t)}{\sqrt{(4\bar{x}^t(1-2\epsilon_2^t)-4)^2+4}} + \eta\frac{4-2\bar{x}^t(1+\epsilon_2^t)}{\sqrt{(-2\bar{x}^t(1+\epsilon_2^t)+4)^2+4}} \geq \eta\frac{\bar{x}_t}{\sqrt{(\bar{x}_t)^2+4}},$$

now using the fact that for Case (3), we have $0.5 \leq \bar{x}^t \leq 1/(1-2\epsilon_2^t)$, we can restate the sufficient condition as

$$\bar{x}^t - 0.5 + 2\eta\frac{4-4\bar{x}^t(1-2\epsilon_2^t)}{\sqrt{(4\bar{x}^t(1-2\epsilon_2^t)-4)^2+4}} + \eta\frac{4-2\bar{x}^t(1+\epsilon_2^t)}{\sqrt{(-2\bar{x}^t(1+\epsilon_2^t)+4)^2+4}} \geq \frac{\eta}{2}, \quad (13)$$

where we have used the fact that $0.5 \leq \bar{x}^t \leq 1.1$ for $\eta < 1/22$ and the fact that the term $\eta\frac{\bar{x}_t}{\sqrt{(\bar{x}_t)^2+4}} > \frac{\eta}{2}$ for $0.5 \leq \bar{x}^t \leq 1.1$. Moreover, $\eta < 1/22$ ensures that $1 + \epsilon_2^t \leq 23/22$. Next, using the fact that $\frac{4-4\bar{x}^t(1-2\epsilon_2^t)}{\sqrt{(4\bar{x}^t(1-2\epsilon_2^t)-4)^2+4}} > 0$ and

$$\frac{4-2\bar{x}^t(1+\epsilon_2^t)}{\sqrt{(-2\bar{x}^t(1+\epsilon_2^t)+4)^2+4}} \geq \frac{4-2\bar{x}^t(23/22)}{\sqrt{(-2\bar{x}^t(1+\epsilon_2^t)+4)^2+4}} \geq \frac{6}{10},$$

Substituting in the l.h.s. of the sufficient condition stated in (13), we get

$$\bar{x}^t - 0.5 + 2\eta\frac{4-4\bar{x}^t(1-2\epsilon_2^t)}{\sqrt{(4\bar{x}^t(1-2\epsilon_2^t)-4)^2+4}} + \eta\frac{4-2\bar{x}^t(1+\epsilon_2^t)}{\sqrt{(-2\bar{x}^t(1+\epsilon_2^t)+4)^2+4}} \geq \frac{6\eta}{10},$$

where we used that fact that $\bar{x}^t \geq 0.5$. Note that $\frac{6\eta}{10} > \frac{\eta}{2}$, therefore, the sufficient condition stated in (13) is satisfied. This further implies that the $\bar{x}^{t+2} \geq 0.5$ during the execution of the algorithm.

Recall that the optimal solution for solving the CO problem is $x^* = 0$. This means Algorithm 1 under both Case I and II fails to converge to the stationary solution.

Hence, the theorem is proved. $\qquad\square$

Finally, we corroborate the result presented in Theorem D.1 via numerical experiment for solving (10) using Case II of Algorithm 1. In Figure 7, we plot the evolution of $\bar{x}^t$ in each communication round. We note that $\bar{x}^t$ is lower bounded by 0.5 as established in the proof of Theorem 4.2 above. In fact, note that for all the settings as the communication rounds increase, $\bar{x}^t$ eventually converges to a quantity that is greater than 1. However, as discussed for the example considered to establish the proof of Theorem 4.1, we know that the true optimizer of the CO problem (10) is $x^* = 0$.

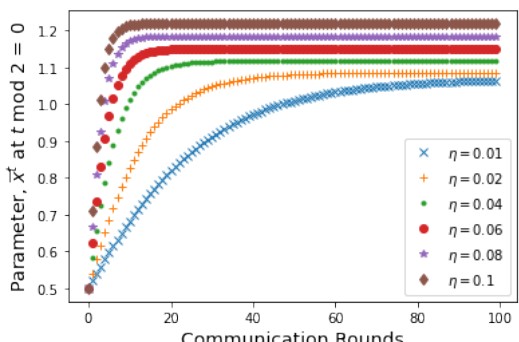

Figure 7: The evolution of parameter $\bar{x}^t$ at each communication round for different choices of step-sizes $\eta$.

# E  PROOF OF THEOREM 4.2

**Theorem E.1** (Modified FedAvg: Convergence for CO). *Suppose we modify Algorithm 1 such that $y_k^t = \bar{y}^t$ is updated at each iteration $t \in \{0, 1, \ldots, T-1\}$ instead of $[t+1 \mod I]$ iterations as in current version of Algorithm 1. Then if functions $f(\cdot)$ and $g_k(x)$ for $k \in [K]$ satisfy Assumptions 3.2, 3.3, and 3.4 such that for a fixed number of local updates $1 \le I \le \mathcal{O}(T^{1/4})$, there exists a choice of $\eta^t > 0$ for $t \in \{0, 1, \ldots, T-1\}$ such that the iterates generated by (modified) Algorithm 1 converge to the stationary point of $\Phi(\cdot)$, where $\Phi(\cdot)$ is defined in (2) with $h(x) = 0$.*

*Proof.* Theorem E.1 is a direct consequence of Theorem 4.3. Therefore, we next prove the main result of the paper in Theorem 4.3. □

# F  PROOF OF THEOREM 4.3

For the purpose of this proof, we define the filtration $\mathcal{F}^t$ as the sigma-algebra generated by the iterates $x_k^1, x_k^1, \ldots, x_k^t$ as

$$\mathcal{F}^t := \sigma(x_k^1, x_k^1, \ldots, x_k^t, \text{ for all } k \in [K]).$$

Moreover, we define the following. Assuming the total training rounds, $T-1$, to be a multiple of $I$, i.e., $T-1 = S \times I$ for some $S \in \mathbb{N}$, we define $t_s := s \times I$ with $s \in \{0, 1, \ldots, S\}$ as the training rounds where the potentially high-dimensional model parameters, $x_k^t$, are shared among the clients. Next, we state Theorem 4.3 again and present the detailed proof of the result.

**Theorem F.1.** *Under Assumptions 3.2, 3.3, and 3.4 and with the choice of step-size $\eta^t = \eta = \sqrt{\frac{|b|K}{T}}$ for all $t \in \{0, 1, \ldots, T-1\}$. Moreover, choosing the momentum parameter $\beta^t = \beta = c_\beta \eta$ where $c_\beta = 4B_g^4 L_f^2$. Then for*

$$T \ge T_{th} := \max\left\{ \frac{4(L_\Phi|b|K + 8B_g^2)^2}{|b|K}, \frac{B_g^4(96L_h^2 + 96B_f^2 L_g^2)^2}{|b|K(L_h^2 + 2B_f^2 L_g^2 + 4B_g^4 L_f^2)^2}, \right.$$

$$\left. (216L_h^2 + 216B_f^2 L_g^2)I^2|b|K \right\}$$

*The iterates generated by Algorithm 2 satisfy*

$$\mathbb{E}\big\|\nabla\Phi(\bar{x}^{a(T)})\big\|^2 \leq \frac{2\big[\Phi(\bar{x}^0) - \Phi(x^*) + \big\|\bar{y}^0 - g(\bar{x}^0)\big\|^2\big]}{\sqrt{|b|KT}} + \frac{K(I-1)^2}{T}\Big[2\bar{L}_{f,g}\sigma_h^2 + 2B_f^2\bar{L}_{f,g}\sigma_g^2\Big]$$

$$+ \frac{1}{\sqrt{|b|KT}}\Big[\big(4L_\Phi + 8B_g^2\big)\sigma_h^2 + \big(4L_\Phi B_f^2 + 4c_\beta^2 + 8B_f^2B_g^2\big)\sigma_g^2\Big]$$

$$+ \frac{|b|K(I-1)^2}{T}\Big[6\bar{L}_{f,g}\Delta_h^2 + 6B_f^2\bar{L}_{f,g}\Delta_g^2\Big] + \frac{1}{\sqrt{|b|KT}}\Big[96B_g^2\,\Delta_h^2 + 96B_f^2B_g^2\,\Delta_g^2\Big].$$

**Corollary F.2.** *Under the same setting as Theorem 4.3, for the choice of local updates $I = T^{1/4}/(|b|K)^{3/4}$, the iterates generated by Algorithm 2 satisfy*

$$\mathbb{E}\big\|\nabla\Phi(\bar{x}^{a(T)})\big\|^2 \leq \frac{2\big[\Phi(\bar{x}^0) - \Phi(x^*) + \big\|\bar{y}^0 - g(\bar{x}^0)\big\|^2\big]}{\sqrt{|b|KT}} + \frac{C_{\sigma_h}}{\sqrt{|b|KT}}\sigma_h^2 + \frac{C_{\sigma_g}}{\sqrt{|b|KT}}\sigma_g^2$$

$$+ \frac{C_{\Delta_h}}{\sqrt{|b|KT}}\Delta_h^2 + \frac{C_{\Delta_g}}{\sqrt{|b|KT}}\Delta_g^2. \qquad (14)$$

*where the constants $C_{\sigma_h}$, $C_{\sigma_g}$, $C_{\Delta_h}$, and $C_{\Delta_g}$ are constants dependent on $L_g$, $L_h$, $L_f$, $B_g$, and $B_f$.*

We prove the Theorem in multiple steps with the help of several intermediate Lemmas.

**Lemma F.3** (**Descent in Function Value**). *Under Assumptions 3.2-3.4, the iterates generated by Algorithm 2 satisfy*

$$\mathbb{E}\big[\Phi(\bar{x}^{t+1}) - \Phi(\bar{x}^t)\big] \leq -\frac{\eta^t}{2}\mathbb{E}\big\|\nabla\Phi(\bar{x}^t)\big\|^2 - \Big(\frac{\eta^t}{2} - (\eta^t)^2L_\Phi\Big)\mathbb{E}\Big\|\frac{1}{K}\sum_{k=1}^K\mathbb{E}\big[\nabla\Phi_k(x_k^t;\bar{\xi}_k^t)\big|\mathcal{F}^t\big]\Big\|^2$$

$$+ \eta^t\big(L_h^2 + 2B_f^2L_g^2 + 4B_g^4L_F^2\big)\frac{1}{K}\sum_{k=1}^K\mathbb{E}\|x_k^t - \bar{x}^t\|^2 + 4B_g^4L_f^2\eta^t\,\mathbb{E}\Big\|\bar{y}^t - \frac{1}{K}\sum_{k=1}^K g_k(x_k^t)\Big\|^2$$

$$+ \frac{2(\eta^t)^2L_\Phi}{K|b_h|}\sigma_h^2 + \frac{2(\eta^t)^2L_\Phi B_f^2}{K|b_g|}\sigma_g^2.$$

*for all $t \in \{0, 1, \ldots, T-1\}$.*

*Proof.* Using the fact that the loss function $\Phi(x)$ is $L_\Phi$-Lipschitz smooth, we get

$$\mathbb{E}\big[\Phi(\bar{x}^{t+1}) - \Phi(\bar{x}^t)\big]$$

$$\leq \mathbb{E}\Big[\langle\nabla\Phi(\bar{x}^t), \bar{x}^{t+1} - \bar{x}^t\rangle + \frac{L_\Phi}{2}\|\bar{x}^{t+1} - \bar{x}^t\|^2\Big]$$

$$\overset{(a)}{\leq} \mathbb{E}\Big[-\eta^t\Big\langle\nabla\Phi(\bar{x}^t), \frac{1}{K}\sum_{k=1}^K\nabla\Phi_k(x_k^t;\bar{\xi}_k^t)\Big\rangle + \frac{(\eta^t)^2L_\Phi}{2}\Big\|\frac{1}{K}\sum_{k=1}^K\nabla\Phi_k(x_k^t;\bar{\xi}_k^t)\Big\|^2\Big]$$

$$\overset{(b)}{\leq} \mathbb{E}\Big[-\eta^t\Big\langle\nabla\Phi(\bar{x}^t), \frac{1}{K}\sum_{k=1}^K\mathbb{E}\big[\nabla\Phi_k(x_k^t;\bar{\xi}_k^t)\big|\mathcal{F}^t\big]\Big\rangle + \frac{(\eta^t)^2L_\Phi}{2}\Big\|\frac{1}{K}\sum_{k=1}^K\nabla\Phi_k(x_k^t;\bar{\xi}_k^t)\Big\|^2\Big]$$

$$\overset{(c)}{\leq} -\frac{\eta^t}{2}\mathbb{E}\big\|\nabla\Phi(\bar{x}^t)\big\|^2 - \Big(\frac{\eta^t}{2} - (\eta^t)^2L_\Phi\Big)\mathbb{E}\Big\|\frac{1}{K}\sum_{k=1}^K\mathbb{E}\big[\nabla\Phi_k(x_k^t;\bar{\xi}_k^t)\big|\mathcal{F}^t\big]\Big\|^2$$

$$+ \frac{\eta^t}{2}\,\underbrace{\mathbb{E}\Big\|\nabla\Phi(\bar{x}^t) - \frac{1}{K}\sum_{k=1}^K\mathbb{E}\big[\nabla\Phi_k(x_k^t;\bar{\xi}_k^t)\big|\mathcal{F}^t\big]\Big\|^2}_{\text{Term I}} \qquad (15)$$

$$+ (\eta^t)^2 L_\Phi \, \mathbb{E} \underbrace{\left\| \frac{1}{K} \sum_{k=1}^K \nabla \Phi_k(x_k^t; \bar{\xi}_k^t) - \frac{1}{K} \sum_{k=1}^K \mathbb{E}\big[\nabla \Phi_k(x_k^t; \bar{\xi}_k^t) \big| \mathcal{F}^t\big] \right\|^2}_{\text{Term II}},$$

where $(a)$ follows from the update step in Algorithm 2; $(b)$ results from moving the conditional expectation w.r.t. the filtration $\mathcal{F}^t$ inside the inner-product; finally, $(c)$ uses the equality $2\langle a, b \rangle = \|a\|^2 + \|b\|^2 - \|a - b\|^2$ for $a, b \in \mathbb{R}^d$ and Lemma C.1 to split the last term.

Next, we consider Terms I and II separately. First, note that from the definition of $\nabla \Phi_k(x_k^t; \bar{\xi}_k^t)$ for all $k \in [K]$, we have

$$\mathbb{E}\big[\nabla \Phi_k(x_k^t; \bar{\xi}_k^t) \big| \mathcal{F}^t\big] = \mathbb{E}\left[ \frac{1}{|b_{h_k}^t|} \sum_{i \in b_{h_k}^t} \nabla h_k(x_k^t; \xi_{k,i}^t) + \frac{1}{|b_{g_k}^t|} \sum_{j \in b_{g_k}^t} \nabla g_k(x_k^t; \zeta_{k,j}^t) \nabla f(\bar{y}^t) \bigg| \mathcal{F}^t \right]$$

$$\overset{(a)}{=} \nabla h_k(x_k^t) + \nabla g_k(x_k^t) \nabla f(\bar{y}^t) \tag{16}$$

where $(a)$ follows from Assumption 3.3. Moreover, from the definition of $\Phi(\bar{x}^t)$, we have

$$\nabla \Phi(\bar{x}^t) = \frac{1}{K} \sum_{k=1}^K \left[ \nabla h_k(\bar{x}^t) + \nabla g_k(\bar{x}^t) \nabla f(g(\bar{x}^t)) \right], \tag{17}$$

where $g(\bar{x}^t) = \frac{1}{K} \sum_{k=1}^K g_k(\bar{x}^t)$. Next, utilizing the expressions obtained in (16) and (17) we bound Term I as

$$\text{Term I} := \mathbb{E}\left\| \nabla \Phi(\bar{x}^t) - \frac{1}{K} \sum_{k=1}^K \mathbb{E}\big[\nabla \Phi_k(x_k^t; \bar{\xi}_k^t) \big| \mathcal{F}^t\big] \right\|^2$$

$$= \mathbb{E}\left\| \frac{1}{K} \sum_{k=1}^K \left[ \nabla h_k(\bar{x}^t) + \nabla g_k(\bar{x}^t) \nabla f(g(\bar{x}^t)) - \big[\nabla h_k(x_k^t) + \nabla g_k(x_k^t) \nabla f(\bar{y}^t)\big] \right] \right\|^2$$

$$\overset{(a)}{\leq} \frac{2}{K} \sum_{k=1}^K \left[ \mathbb{E}\|\nabla h_k(x_k^t) - \nabla h_k(\bar{x}^t)\|^2 + \|\nabla g_k(x_k^t) \nabla f(\bar{y}^t) - \nabla g_k(\bar{x}^t) \nabla f(g(\bar{x}^t))\|^2 \right]$$

$$\overset{(b)}{\leq} \frac{2L_h^2}{K} \sum_{k=1}^K \mathbb{E}\|x_k^t - \bar{x}^t\|^2 + \frac{4}{K} \sum_{k=1}^K \mathbb{E}\big\| \nabla g_k(x_k^t) \big[\nabla f(\bar{y}^t) - \nabla f(g(\bar{x}^t))\big] \big\|^2$$

$$+ \frac{4}{K} \sum_{k=1}^K \mathbb{E}\big\| \big[\nabla g_k(x_k^t) - \nabla g_k(\bar{x}^t)\big] \nabla f(g(\bar{x}^t)) \big\|^2$$

$$\overset{(c)}{\leq} \frac{2L_h^2}{K} \sum_{k=1}^K \mathbb{E}\|x_k^t - \bar{x}^t\|^2 + \frac{4B_g^2}{K} \sum_{k=1}^K \mathbb{E}\big\| \nabla f(\bar{y}^t) - \nabla f(g(\bar{x}^t)) \big\|^2$$

$$+ \frac{4B_f^2}{K} \sum_{k=1}^K \mathbb{E}\|\nabla g_k(x_k^t) - \nabla g_k(\bar{x}^t)\|^2$$

$$\overset{(d)}{\leq} \left( \frac{2L_h^2}{K} + \frac{4B_f^2 L_g^2}{K} \right) \sum_{k=1}^K \mathbb{E}\|x_k^t - \bar{x}^t\|^2 + 4B_g^2 L_f^2 \underbrace{\mathbb{E}\|\bar{y}^t - g(\bar{x}^t)\|^2}_{\text{Term III}}.$$

Next, let us consider Term III above.

$$\text{Term III} := \mathbb{E}\big\|\bar{y}^t - g(\bar{x}^t)\big\|^2$$

$$\overset{(a)}{\leq} 2\mathbb{E}\Big\|\bar{y}^t - \frac{1}{K}\sum_{k=1}^{K} g_k(x_k^t)\Big\|^2 + 2\mathbb{E}\Big\|\frac{1}{K}\sum_{k=1}^{K} g_k(x_k^t) - g(\bar{x}^t)\Big\|^2$$

$$\overset{(b)}{\leq} 2\mathbb{E}\Big\|\bar{y}^t - \frac{1}{K}\sum_{k=1}^{K} g_k(x_k^t)\Big\|^2 + \frac{2}{K}\sum_{k=1}^{K}\mathbb{E}\big\|g_k(x_k^t) - g_k(\bar{x}^t)\big\|^2$$

$$\overset{(c)}{\leq} 2\mathbb{E}\Big\|\bar{y}^t - \frac{1}{K}\sum_{k=1}^{K} g_k(x_k^t)\Big\|^2 + \frac{2B_g^2}{K}\sum_{k=1}^{K}\mathbb{E}\|x_k^t - \bar{x}^t\|^2,$$

where $(a)$ follows from the application of Lemma C.1; $(b)$ results from the definition of $g(x) = \frac{1}{K}\sum_{k=1}^{K} g_k(x)$ and the use of Lemma C.1; finally $(c)$ results from the Lipschitz-ness of $g_k(\cdot)$ for all $k \in [K]$.

Next, we consider Term II below

$$\text{Term II} := \mathbb{E}\Big\|\frac{1}{K}\sum_{k=1}^{K}\nabla\Phi_k(x_k^t;\bar{\xi}_k^t) - \frac{1}{K}\sum_{k=1}^{K}\mathbb{E}\big[\nabla\Phi_k(x_k^t;\bar{\xi}_k^t)\big|\mathcal{F}^t\big]\Big\|^2$$

$$\overset{(a)}{=} \frac{1}{K^2}\sum_{k=1}^{K}\mathbb{E}\big\|\nabla\Phi_k(x_k^t;\bar{\xi}_k^t) - \mathbb{E}\big[\nabla\Phi_k(x_k^t;\bar{\xi}_k^t)\big|\mathcal{F}^t\big]\big\|^2$$

$$\overset{(b)}{=} \frac{1}{K^2}\sum_{k=1}^{K}\mathbb{E}\Big\|\frac{1}{|b_{h_k}^t|}\sum_{i\in b_{h_k}^t}\nabla h_k(x_k^t;\xi_{k,i}^t) + \frac{1}{|b_{g_k}^t|}\sum_{j\in b_{g_k}^t}\nabla g_k(x_k^t;\zeta_{k,j}^t)\nabla f(\bar{y}^t)$$

$$- \Big[\nabla h_k(x_k^t) + \nabla g_k(x_k^t)\nabla f(\bar{y}^t)\Big]\Big\|^2$$

$$\overset{(c)}{=} \frac{2}{K^2}\sum_{k=1}^{K}\mathbb{E}\Big\|\frac{1}{|b_{h_k}^t|}\sum_{i\in b_{h_k}^t}\nabla h_k(x_k^t;\xi_{k,i}^t) - \nabla h_k(x_k^t)\Big\|^2$$

$$+ \frac{2}{K^2}\sum_{k=1}^{K}\mathbb{E}\Big\|\frac{1}{|b_{g_k}^t|}\sum_{j\in b_{g_k}^t}\nabla g_k(x_k^t;\zeta_{k,j}^t)\nabla f(\bar{y}^t) - \nabla g_k(x_k^t)\nabla f(\bar{y}^t)\Big\|^2$$

$$\overset{(d)}{\leq} \frac{2\sigma_h^2}{K|b_h|} + \frac{2\sigma_g^2 B_f^2}{K|b_g|},$$

where $(a)$ follows from the application of Lemma C.1; $(b)$ follows from the definition of the stochastic gradient in (7) and its expectation in (16); $(c)$ again uses Lemma C.1; Finally, $(d)$ uses Cauchy-Schwartz inequality, Lipschitzness of $f(\bar{y}^t)$ and Assumption 3.3 and using $|b_{h_k}| = |b_h|$ and $|b_{g_k}| = |b_g|$ for all $k \in [K]$.

Next, substituting the upper bounds obtained for Terms I, II, and III into (15), we get

$$\mathbb{E}\big[\Phi(\bar{x}^{t+1}) - \Phi(\bar{x}^t)\big] \leq -\frac{\eta^t}{2}\mathbb{E}\big\|\nabla\Phi(\bar{x}^t)\big\|^2 - \Big(\frac{\eta^t}{2} - (\eta^t)^2 L_\Phi\Big)\mathbb{E}\Big\|\frac{1}{K}\sum_{k=1}^{K}\mathbb{E}\big[\nabla\Phi_k(x_k^t;\bar{\xi}_k^t)\big|\mathcal{F}^t\big]\Big\|^2$$

$$+ \eta^t\big(L_h^2 + 2B_f^2 L_g^2 + 4B_g^4 L_F^2\big)\underbrace{\frac{1}{K}\sum_{k=1}^{K}\mathbb{E}\|x_k^t - \bar{x}^t\|^2}_{\text{Term IV}} + 4B_g^4 L_f^2\eta^t\underbrace{\mathbb{E}\Big\|\bar{y}^t - \frac{1}{K}\sum_{k=1}^{K}g_k(x_k^t)\Big\|^2}_{\text{Term V}}$$

$$+ \frac{2(\eta^t)^2 L_\Phi}{K|b_h|}\sigma_h^2 + \frac{2(\eta^t)^2 L_\Phi B_f^2}{K|b_g|}\sigma_g^2. \tag{18}$$

Therefore, we have the proof of the Lemma. $\qquad\square$

Next, we bound Terms IV and V in (18) in the next Lemmas. Let us first consider Term IV.

**Lemma F.4** (**Client Drift**). *Under Assumptions 3.2-3.4, the iterates generated by Algorithm 2 satisfy*

$$\frac{1}{K}\sum_{k=1}^{K}\mathbb{E}\|x_k^t - \bar{x}^t\| \leq (I-1)\Big(24L_h^2 + 24B_f^2L_g^2\Big)\sum_{\ell=t_s}^{t-1}\frac{(\eta^\ell)^2}{K}\sum_{k=1}^{K}\mathbb{E}\|x_k^\ell - \bar{x}^\ell\|^2$$

$$+ (I-1)\Big(\frac{4}{|b_h^t|}\sigma_h^2 + \frac{4B_f^2}{|b_g^t|}\sigma_g^2\Big)\sum_{\ell=t_s}^{t-1}(\eta^\ell)^2 + (I-1)\Big(12\Delta_h^2 + 12B_f^2\Delta_g^2\Big)\sum_{\ell=t_s}^{t-1}(\eta^\ell)^2.$$

*Proof.* Recall from the definition of $t_s$ that we have $x_k^{t_s} = \bar{x}^{t_s}$ for all $s \in \{0, 1, \ldots, S\}$. Next, we have from the update rule in Algorithm 2 that for all $t \in [t_s + 1, t_{s+1} - 1]$

$$x_k^t = x_k^{t-1} - \eta^{t-1}\nabla\Phi_k(x_k^{t-1}; \bar{\xi}_k^{t-1}) \overset{(a)}{=} x_k^{t_s} - \sum_{\ell=t_s}^{t-1}\eta^\ell\nabla\Phi_k(x_k^\ell; \bar{\xi}_k^\ell). \tag{19}$$

where $(a)$ results from unrolling the updates from Algorithm 2. Similarly, we have

$$\bar{x}^t = \bar{x}^{t-1} - \eta^{t-1}\frac{1}{K}\sum_{k=1}^{K}\nabla\Phi_k(x_k^{t-1}; \bar{\xi}_k^{t-1}) = \bar{x}^{t_s} - \frac{1}{K}\sum_{k=1}^{K}\sum_{\ell=t_s}^{t-1}\eta^\ell\nabla\Phi_k(x_k^\ell; \bar{\xi}_k^\ell) \tag{20}$$

Bounding Term IV, we have

$$\text{Term IV} := \frac{1}{K}\sum_{k=1}^{K}\mathbb{E}\|x_k^t - \bar{x}^t\|^2$$

$$\overset{(a)}{=} \frac{1}{K}\sum_{k=1}^{K}\mathbb{E}\Big\|\sum_{\ell=t_s}^{t-1}\eta^\ell\nabla\Phi_k(x_k^\ell; \bar{\xi}_k^\ell) - \frac{1}{K}\sum_{k=1}^{K}\sum_{\ell=t_s}^{t-1}\eta^\ell\nabla\Phi_k(x_k^\ell; \bar{\xi}_k^\ell)\Big\|^2$$

$$\overset{(b)}{=} (I-1)\sum_{\ell=t_s}^{t-1}\frac{(\eta^\ell)^2}{K}\sum_{k=1}^{K}\underbrace{\mathbb{E}\Big\|\nabla\Phi_k(x_k^\ell; \bar{\xi}_k^\ell) - \frac{1}{K}\sum_{k=1}^{K}\nabla\Phi_k(x_k^\ell; \bar{\xi}_k^\ell)\Big\|^2}_{\text{Term VI}}$$

where $(a)$ follows from (19) and (20) and $(b)$ follows from the application of Lemma C.1.

Next, we bound Term VI in the above expression.

$$\text{Term VI} := \mathbb{E}\Big\|\nabla\Phi_k(x_k^\ell; \bar{\xi}_k^\ell) - \frac{1}{K}\sum_{k=1}^{K}\nabla\Phi_k(x_k^\ell; \bar{\xi}_k^\ell)\Big\|^2$$

$$\overset{(a)}{=} \mathbb{E}\Big\|\frac{1}{|b_{h_k}^\ell|}\sum_{i \in b_{h_k}^\ell}\nabla h_k(x_k^\ell; \xi_{k,i}^\ell) + \frac{1}{|b_{g_k}^\ell|}\sum_{j \in b_{g_k}^\ell}\nabla g_k(x_k^\ell; \zeta_{k,j}^\ell)\nabla f(\bar{y}^\ell)$$

$$- \frac{1}{K}\sum_{k=1}^{K}\Big[\frac{1}{|b_{h_k}^\ell|}\sum_{i \in b_{h_k}^\ell}\nabla h_k(x_k^\ell; \xi_{k,i}^\ell) + \frac{1}{|b_{g_k}^\ell|}\sum_{j \in b_{g_k}^\ell}\nabla g_k(x_k^\ell; \zeta_{k,j}^\ell)\nabla f(\bar{y}^\ell)\Big]\Big\|^2$$

$$\overset{(b)}{\leq} 2\mathbb{E}\Big\|\frac{1}{|b_{h_k}^\ell|}\sum_{i \in b_{h_k}^\ell}\nabla h_k(x_k^\ell; \xi_{k,i}^\ell) - \frac{1}{K}\sum_{k=1}^{K}\frac{1}{|b_{h_k}^\ell|}\sum_{i \in b_{h_k}^\ell}\nabla h_k(x_k^\ell; \xi_{k,i}^\ell)\Big\|^2$$

$$+ 2\mathbb{E}\Big\|\frac{1}{|b_{g_k}^\ell|}\sum_{j \in b_{g_k}^\ell}\nabla g_k(x_k^\ell; \zeta_{k,j}^\ell)\nabla f(\bar{y}^\ell) - \frac{1}{K}\sum_{k=1}^{K}\frac{1}{|b_{g_k}^\ell|}\sum_{j \in b_{g_k}^\ell}\nabla g_k(x_k^\ell; \zeta_{k,j}^t)\nabla f(\bar{y}^\ell)\Big\|^2$$

$$\overset{(c)}{\leq} 2\,\mathbb{E}\left\|\frac{1}{|b^\ell_{h_k}|}\sum_{i\in b^\ell_{h_k}}\nabla h_k(x^\ell_k;\xi^\ell_{k,i}) - \frac{1}{K}\sum_{k=1}^K\frac{1}{|b^\ell_{h_k}|}\sum_{i\in b^\ell_{h_k}}\nabla h_k(x^\ell_k;\xi^\ell_{k,i})\right\|^2}_{\text{Term VII}}$$

$$+\,2B_f^2\,\underbrace{\mathbb{E}\left\|\frac{1}{|b^\ell_{g_k}|}\sum_{j\in b^\ell_{g_k}}\nabla g_k(x^\ell_k;\zeta^\ell_{k,j}) - \frac{1}{K}\sum_{k=1}^K\frac{1}{|b^\ell_{g_k}|}\sum_{j\in b^\ell_{g_k}}\nabla g_k(x^\ell_k;\zeta^\ell_{k,j})\right\|^2}_{\text{Term VIII}},$$

where $(a)$ results from the definition of the stochastic gradient evaluated in (7); $(b)$ uses Lemma C.1; and $(c)$ utilizes the Cauchy-Schwartz inequality combined with the Lipschitzness of $f(\cdot)$. Next, in order to upper bound Term VI, we bound Terms VII and VIII separately. First, let us consider Term VII above

$$\text{Term VII} := \mathbb{E}\left\|\frac{1}{|b^\ell_{h_k}|}\sum_{i\in b^\ell_{h_k}}\nabla h_k(x^\ell_k;\xi^\ell_{k,i}) - \frac{1}{K}\sum_{k=1}^K\frac{1}{|b^\ell_{h_k}|}\sum_{i\in b^\ell_{h_k}}\nabla h_k(x^\ell_k;\xi^\ell_{k,i})\right\|^2$$

$$\overset{(a)}{\leq} 2\mathbb{E}\left\|\left[\frac{1}{|b^\ell_{h_k}|}\sum_{i\in b^\ell_{h_k}}\nabla h_k(x^\ell_k;\xi^\ell_{k,i}) - \nabla h_k(x^\ell_k)\right] - \frac{1}{K}\sum_{k=1}^K\left[\frac{1}{|b^\ell_{h_k}|}\sum_{i\in b^\ell_{h_k}}\nabla h_k(x^\ell_k;\xi^\ell_{k,i}) - \nabla h_k(x^\ell_k)\right]\right\|^2$$

$$+\,2\mathbb{E}\left\|\nabla h_k(x^\ell_k) - \frac{1}{K}\sum_{k=1}^K\nabla h_k(x^\ell_k)\right\|^2$$

$$\overset{(b)}{\leq} 2\mathbb{E}\left\|\frac{1}{|b^\ell_{h_k}|}\sum_{i\in b^\ell_{h_k}}\nabla h_k(x^\ell_k;\xi^\ell_{k,i}) - \nabla h_k(x^\ell_k)\right\|^2 + 2\mathbb{E}\left\|\nabla h_k(x^\ell_k) - \frac{1}{K}\sum_{k=1}^K\nabla h_k(x^\ell_k)\right\|^2$$

$$\overset{(c)}{\leq} \frac{2\sigma_h^2}{|b^\ell_{h_k}|} + 2\,\underbrace{\mathbb{E}\left\|\nabla h_k(x^\ell_k) - \frac{1}{K}\sum_{k=1}^K\nabla h_k(x^\ell_k)\right\|^2}_{\text{Term IX}},$$

where $(a)$ utilizes Lemma C.1; $(b)$ results from the application of Lemma C.2; and $(c)$ results from Assumption 3.3.

Next, we bound Term IX below

$$\text{Term IX} := \mathbb{E}\left\|\nabla h_k(x^\ell_k) - \frac{1}{K}\sum_{k=1}^K\nabla h_k(x^\ell_k)\right\|^2$$

$$\overset{(a)}{\leq} 3\mathbb{E}\left\|\nabla h_k(x^\ell_k) - \nabla h_k(\bar{x}^\ell)\right\|^2 + 3\mathbb{E}\left\|\frac{1}{K}\sum_{k=1}^K\left[\nabla h_k(\bar{x}^\ell) - \nabla h_k(x^\ell_k)\right]\right\|^2$$

$$+\,3\mathbb{E}\left\|\nabla h_k(\bar{x}^\ell) - \frac{1}{K}\sum_{k=1}^K\nabla h_k(\bar{x}^\ell)\right\|^2$$

$$\overset{(b)}{\leq} 3L_h^2\mathbb{E}\left\|x^\ell_k - \bar{x}^\ell\right\|^2 + \frac{3L_h^2}{K}\sum_{k=1}^K\mathbb{E}\left\|x^\ell_k - \bar{x}^\ell\right\|^2 + 3\mathbb{E}\left\|\nabla h_k(\bar{x}^\ell) - \nabla h(\bar{x}^\ell)\right\|^2$$

$$\overset{(c)}{\leq} 3L_h^2\mathbb{E}\left\|x^\ell_k - \bar{x}^\ell\right\|^2 + \frac{3L_h^2}{K}\sum_{k=1}^K\mathbb{E}\left\|x^\ell_k - \bar{x}^\ell\right\|^2 + 3\Delta_h^2,$$

where $(a)$ results from the application of Lemma C.1; $(b)$ utilizes Lipschitz smoothness of $h(\cdot)$ and the definition of $h(x) = \frac{1}{K}\sum_{k=1}^K h_k(x)$; finally, $(c)$ results from the bounded heterogeneity

assumption Assumption 3.4. Substituting the bound on Term IX in the bound of Term VII, we get

$$\text{Term VII} \le \frac{2\sigma_h^2}{|b_{h_k}^t|} + 6L_h^2\mathbb{E}\|x_k^\ell - \bar{x}^\ell\|^2 + \frac{6L_h^2}{K}\sum_{k=1}^K \mathbb{E}\|x_k^\ell - \bar{x}^\ell\|^2 + 6\Delta_h^2.$$

Similarly, we bound Term VIII as

$$\text{Term VIII} := \mathbb{E}\left\|\frac{1}{|b_{g_k}^\ell|}\sum_{j\in b_{g_k}^\ell}\nabla g_k(x_k^\ell; \zeta_{k,j}^\ell) - \frac{1}{K}\sum_{k=1}^K \frac{1}{|b_{g_k}^\ell|}\sum_{j\in b_{g_k}^\ell}\nabla g_k(x_k^\ell; \zeta_{k,j}^\ell)\right\|^2$$

$$\overset{(a)}{\le} 2\mathbb{E}\left\|\left[\frac{1}{|b_{g_k}^\ell|}\sum_{i\in b_{g_k}^\ell}\nabla g_k(x_k^\ell; \zeta_{k,i}^\ell) - \nabla g_k(x_k^\ell)\right] - \frac{1}{K}\sum_{k=1}^K\left[\frac{1}{|b_{g_k}^\ell|}\sum_{i\in b_{g_k}^\ell}\nabla g_k(x_k^\ell; \zeta_{k,i}^\ell) - \nabla g_k(x_k^\ell)\right]\right\|^2$$

$$+ 2\mathbb{E}\left\|\nabla g_k(x_k^\ell) - \frac{1}{K}\sum_{k=1}^K\nabla g_k(x_k^\ell)\right\|^2$$

$$\overset{(b)}{\le} 2\mathbb{E}\left\|\frac{1}{|b_{g_k}^\ell|}\sum_{i\in b_{g_k}^\ell}\nabla g_k(x_k^\ell; \zeta_{k,i}^\ell) - \nabla g_k(x_k^\ell)\right\|^2 + 2\mathbb{E}\left\|\nabla g_k(x_k^\ell) - \frac{1}{K}\sum_{k=1}^K\nabla g_k(x_k^\ell)\right\|^2$$

$$\overset{(c)}{\le} \frac{2\sigma_g^2}{|b_{g_k}^\ell|} + 2\underbrace{\mathbb{E}\left\|\nabla g_k(x_k^\ell) - \frac{1}{K}\sum_{k=1}^K\nabla g_k(x_k^\ell)\right\|^2}_{\text{Term X}},$$

where $(a)$ utilizes Lemma C.1; $(b)$ results from the application of Lemma C.2; and $(c)$ results from Assumption 3.3. Next, we bound Term X below

$$\text{Term X} := \mathbb{E}\left\|\nabla g_k(x_k^\ell) - \frac{1}{K}\sum_{k=1}^K\nabla g_k(x_k^\ell)\right\|^2$$

$$\overset{(a)}{\le} 3\mathbb{E}\|\nabla g_k(x_k^\ell) - \nabla g_k(\bar{x}^\ell)\|^2 + 3\mathbb{E}\left\|\frac{1}{K}\sum_{k=1}^K\left[\nabla g_k(\bar{x}^\ell) - \nabla g_k(x_k^\ell)\right]\right\|^2$$

$$+ 3\mathbb{E}\left\|\nabla g_k(\bar{x}^\ell) - \frac{1}{K}\sum_{k=1}^K\nabla g_k(\bar{x}^\ell)\right\|^2$$

$$\overset{(b)}{\le} 3L_g^2\mathbb{E}\|x_k^\ell - \bar{x}^\ell\|^2 + \frac{3L_g^2}{K}\sum_{k=1}^K \mathbb{E}\|x_k^\ell - \bar{x}^\ell\|^2 + 3\mathbb{E}\|\nabla g_k(\bar{x}^\ell) - \nabla g(\bar{x}^\ell)\|^2$$

$$\overset{(c)}{\le} 3L_g^2\mathbb{E}\|x_k^\ell - \bar{x}^\ell\|^2 + \frac{3L_g^2}{K}\sum_{k=1}^K \mathbb{E}\|x_k^\ell - \bar{x}^\ell\|^2 + 3\Delta_g^2,$$

where $(a)$ results from the application of Lemma C.1; $(b)$ utilizes Lipschitz smoothness of $g(\cdot)$ and the definition of $g(x) = \frac{1}{K}\sum_{k=1}^K g_k(x)$; finally, $(c)$ results from the bounded heterogeneity assumption Assumption 3.4. Substituting the bound on Term X in the bound of Term VIII, we get

$$\text{Term VIII} \le \frac{2\sigma_g^2}{|b_{g_k}^\ell|} + 6L_g^2\mathbb{E}\|x_k^\ell - \bar{x}^\ell\|^2 + \frac{6L_g^2}{K}\sum_{k=1}^K \mathbb{E}\|x_k^\ell - \bar{x}^\ell\|^2 + 6\Delta_g^2.$$

Next, we substitute the upper bounds on Terms VII and VIII in the expression of Term VI, we get

$$
\text{Term VI} \leq \frac{4}{|b_{h_k}^\ell|}\sigma_h^2 + 12L_h^2 \mathbb{E}\|x_k^\ell - \bar{x}^\ell\|^2 + \frac{12L_h^2}{K}\sum_{k=1}^K \mathbb{E}\|x_k^\ell - \bar{x}^\ell\|^2 + 12\Delta_h^2
$$

$$
+ \frac{4B_f^2}{|b_{g_k}^\ell|}\sigma_g^2 + 12B_f^2 L_g^2 \mathbb{E}\|x_k^\ell - \bar{x}^\ell\|^2 + \frac{12B_f^2 L_g^2}{K}\sum_{k=1}^K \mathbb{E}\|x_k^\ell - \bar{x}^\ell\|^2 + 12B_f^2 \Delta_g^2
$$

$$
= \left(12L_h^2 + 12B_f^2 L_g^2\right)\mathbb{E}\|x_k^\ell - \bar{x}^\ell\|^2 + \left(\frac{12L_h^2 + 12B_f^2 L_g^2}{K}\right)\sum_{k=1}^K \mathbb{E}\|x_k^\ell - \bar{x}^\ell\|^2
$$

$$
+ \frac{4}{|b_{h_k}^\ell|}\sigma_h^2 + \frac{4B_f^2}{|b_{g_k}^\ell|}\sigma_g^2 + 12\Delta_h^2 + 12B_f^2 \Delta_g^2.
$$

Therefore, we finally have the bound on Term IV as

$$
\text{Term IV} \leq (I-1)\left(24L_h^2 + 24B_f^2 L_g^2\right)\sum_{\ell=t_s}^{t-1}\frac{(\eta^\ell)^2}{K}\sum_{k=1}^K \mathbb{E}\|x_k^\ell - \bar{x}^\ell\|^2
$$

$$
+ (I-1)\left(\frac{4}{|b_h^t|}\sigma_h^2 + \frac{4B_f^2}{|b_g^t|}\sigma_g^2\right)\sum_{\ell=t_s}^{t-1}(\eta^\ell)^2 + (I-1)\left(12\Delta_h^2 + 12B_f^2 \Delta_g^2\right)\sum_{\ell=t_s}^{t-1}(\eta^\ell)^2.
$$

where we have chosen $|b_{h_k}^\ell| = |b_h^t|$ and $|b_{g_k}^\ell| = |b_g^t|$ for all $k \in [K]$ and $\ell \in \{0, \ldots, T-1\}$.

Therefore, we have proof of the Lemma. $\qquad\square$

Next, we bound Term V from (18), we have

**Lemma F.5** (**Descent in the estimate of** $g(x)$)**.** *Under Assumptions 3.2-3.4, the iterates generated by Algorithm 2 satisfy:*

$$
\mathbb{E}\left\|\bar{y}^t - \frac{1}{K}\sum_{k=1}^K g_k(x_k^t)\right\|^2
$$

$$
\leq (1-\beta^t)^2 \mathbb{E}\left\|\bar{y}^{t-1} - \frac{1}{K}\sum_{k=1}^K g_k(x_k^{t-1})\right\|^2 + \frac{8(\eta^t)^2(1-\beta^t)^2 B_g^2}{|b_g|K}\mathbb{E}\left\|\frac{1}{K}\sum_{k=1}^K \mathbb{E}[\Phi_k(x_k^t;\bar{\xi}_k^t)|\mathcal{F}^t]\right\|^2
$$

$$
+ \frac{(\eta^t)^2(1-\beta^t)^2 B_g^2(96L_h^2 + 96B_f^2 L_g^2)}{|b_g|K^2}\sum_{k=1}^K \mathbb{E}\|x_k^t - \bar{x}^t\|^2 + \frac{4(\eta^t)^2(1-\beta^t)^2 B_g^2}{|b_h|K}\sigma_h^2
$$

$$
+ \frac{2(\beta^t)^2 + 4(\eta^t)^2(1-\beta^t)^2 B_g^2 B_f^2}{|b_g|K}\sigma_g^2 + \frac{48(\eta^t)^2(1-\beta^t)^2 B_g^2}{|b_g|K}\Delta_h^2 + \frac{48(\eta^t)^2(1-\beta^t)^2 B_f^2 B_g^2}{|b_g|K}\Delta_g^2.
$$

*where we have chosen $|b_h^t| = |b_h|$ and $|b_{g_k}^t| = |b_g|$ for all $k \in [K]$ and $t \in [T]$.*

*Proof.* From the definition of Term V, we have

$$
\text{Term V} := \mathbb{E}\left\|\bar{y}^{t+1} - \frac{1}{K}\sum_{k=1}^K g_k(x_k^{t+1})\right\|^2
$$

$$
\overset{(a)}{=} \mathbb{E}\left\|\frac{1}{K}\sum_{k=1}^K \left[y_k^{t+1} - g_k(x_k^{t+1})\right]\right\|^2
$$

$$
\overset{(b)}{=} \mathbb{E}\left\|\frac{1}{K}\sum_{k=1}^K \left[(1-\beta^{t+1})\left(y_k^t + \frac{1}{|b_{g_k}^{t+1}|}\sum_{i\in b_{g_k}^{t+1}} g_k(x_k^{t+1};\zeta_{k,i}^{t+1}) - \frac{1}{|b_{g_k}^{t+1}|}\sum_{i\in b_{g_k}^{t+1}} g_k(x_k^t;\zeta_{k,i}^{t+1})\right)\right.\right.
$$

$$
\left.\left. + \frac{\beta^{t+1}}{|b_{g_k}^{t+1}|}\sum_{i\in b_{g_k}^{t+1}} g_k(x_k^{t+1},\zeta_{k,i}^{t+1}) - g_k(x_k^{t+1})\right]\right\|^2
$$

$$\stackrel{(c)}{=} (1-\beta^{t+1})^2\, \mathbb{E}\left\|\frac{1}{K}\sum_{k=1}^{K}\left[y_k^t - g_k(x_k^t)\right]\right\|^2$$

$$+\mathbb{E}\left\|\frac{1}{K}\sum_{k=1}^{K}\left[(1-\beta^{t+1})\left[(g_k(x_k^t) - g_k(x_k^{t+1})) - \frac{1}{|b_{g_k}^{t+1}|}\sum_{i\in b_{g_k}^{t+1}}\left(g_k(x_k^t;\zeta_{k,i}^{t+1}) - g_k(x_k^{t+1};\zeta_{k,i}^{t+1})\right)\right]\right.\right.$$

$$\left.\left.+\beta^{t+1}\left(\frac{1}{|b_{g_k}^{t+1}|}\sum_{i\in b_{g_k}^{t+1}}g_k(x_k^{t+1};\zeta_{k,i}^{t+1}) - g_k(x_k^{t+1})\right)\right]\right\|^2$$

$$\stackrel{(d)}{\leq} (1-\beta^{t+1})^2\, \mathbb{E}\left\|\bar{y}^t - \frac{1}{K}\sum_{k=1}^{K}g_k(x_k^t)\right\|^2 + \frac{2(\beta^{t+1})^2}{|b_g|K}\sigma_g^2$$

$$+\frac{2(1-\beta^{t+1})^2}{K^2}\sum_{k=1}^{K}\frac{1}{|b_g|^2}\sum_{i\in b_{g_k}^{t+1}}\mathbb{E}\left\|(g_k(x_k^t) - g(x_k^{t+1})) - (g_k(x_k^t;\zeta_{k,i}^{t+1}) - g_k(x_k^{t+1};\zeta_{k,i}^{t+1}))\right\|^2$$

$$\stackrel{(e)}{\leq} (1-\beta^{t+1})^2\, \mathbb{E}\left\|\bar{y}^t - \frac{1}{K}\sum_{k=1}^{K}g_k(x_k^t)\right\|^2 + \frac{2(\beta^{t+1})^2}{|b_g|K}\sigma_g^2$$

$$+\frac{2(1-\beta^{t+1})^2}{K^2}\sum_{k=1}^{K}\frac{1}{|b_g|^2}\sum_{i\in b_{g_k}^{t+1}}\mathbb{E}\left\|g_k(x_k^t;\zeta_{k,i}^{t+1}) - g_k(x_k^{t+1};\zeta_{k,i}^{t+1})\right\|^2$$

$$\stackrel{(f)}{\leq} (1-\beta^{t+1})^2\, \mathbb{E}\left\|\bar{y}^t - \frac{1}{K}\sum_{k=1}^{K}g_k(x_k^t)\right\|^2 + \frac{2(\beta^{t+1})^2}{|b_g|K}\sigma_g^2 + \frac{2(1-\beta^{t+1})^2 B_g^2}{|b_g|K^2}\sum_{k=1}^{K}\mathbb{E}\left\|x_k^{t+1} - x_k^t\right\|^2$$

$$\stackrel{(g)}{\leq} (1-\beta^{t+1})^2\, \mathbb{E}\left\|\bar{y}^t - \frac{1}{K}\sum_{k=1}^{K}g_k(x_k^t)\right\|^2 + \frac{2(\beta^{t+1})^2}{|b_g|K}\sigma_g^2$$

$$+\frac{2(\eta^t)^2(1-\beta^{t+1})^2 B_g^2}{|b_g|K^2}\sum_{k=1}^{K}\underbrace{\mathbb{E}\left\|\nabla\Phi_k(x_k^t;\bar{\xi}_k^t)\right\|^2}_{\text{Term XI}},$$

where $(a)$ follows from the definition of $\bar{y}^{t+1}$; $(b)$ uses the update rule (8) for $y_k^{t+1}$; $(c)$ results from adding and subtracting $(1-\beta^{t+1})g_k(x_k^t)$ and utilizing the fact that the second term in the expression has zero-mean which follows from Assumption 3.3; $(d)$ uses Young's inequality, Assumption 3.3 and by choosing $|b_h^t| = |b_h|$ and $|b_{g_k}^t| = |b_g|$ for all $k\in[K]$ and $t\in[T]$; $(e)$ results from the fact that for a random variable $X$, we have $\mathbb{E}\|X - \mathbb{E}[X]\|^2 \leq \mathbb{E}\|X\|^2$; $(f)$ uses the mean-squared Lipschitzness of $g_k(\cdot)$ in Assumption 3.2; finally $(g)$ results from the update rule of Algorithm 2.

Next, we bound Term XI below

$$\text{Term XI} := \mathbb{E}\left\|\nabla\Phi_k(x_k^t;\bar{\xi}_k^t)\right\|^2$$

$$\stackrel{(a)}{\leq} 2\mathbb{E}\left\|\nabla\Phi_k(x_k^t;\bar{\xi}_k^t) - \mathbb{E}[\nabla\Phi_k(x_k^t;\bar{\xi}_k^t)|\mathcal{F}^t]\right\|^2 + 2\mathbb{E}\left\|\mathbb{E}[\nabla\Phi_k(x_k^t;\bar{\xi}_k^t)|\mathcal{F}^t]\right\|^2$$

$$\stackrel{(b)}{\leq} \frac{2\sigma_h^2}{|b_h|} + \frac{2\sigma_g^2 B_f^2}{|b_g|} + 4\,\mathbb{E}\underbrace{\left\|\mathbb{E}[\nabla\Phi_k(x_k^t;\bar{\xi}_k^t)|\mathcal{F}^t] - \frac{1}{K}\sum_{k=1}^{K}\mathbb{E}[\nabla\Phi_k(x_k^t;\bar{\xi}_k^t)|\mathcal{F}^t]\right\|^2}_{\text{Term XII}}$$

$$+ 4\mathbb{E}\left\|\frac{1}{K}\sum_{k=1}^{K}\mathbb{E}[\nabla\Phi_k(x_k^t;\bar{\xi}_k^t)|\mathcal{F}^t]\right\|^2,$$

where $(a)$ results from the application of Young's inequality and $(b)$ results from Assumptions 3.2 and 3.3 along with the application of Young's inequality.

Next, we bound Term XII in the above expression.

$$
\text{Term XII} := \mathbb{E}\left\|\mathbb{E}[\nabla\Phi_k(x_k^t;\bar{\xi}_k^t)|\mathcal{F}^t] - \frac{1}{K}\sum_{k=1}^{K}\mathbb{E}[\nabla\Phi_k(x_k^t;\bar{\xi}_k^t)|\mathcal{F}^t]\right\|^2
$$

$$
\stackrel{(a)}{=} \mathbb{E}\left\|\nabla h_k(x_k^t) + \nabla g_k(x_k^t)\nabla f(\bar{y}^t) - \left[\frac{1}{K}\sum_{k=1}^{K}\left(\nabla h_k(x_k^t) + \nabla g_k(x_k^t)\nabla f(\bar{y}^t)\right)\right]\right\|^2
$$

$$
\stackrel{(b)}{\leq} 2\mathbb{E}\left\|\nabla h_k(x_k^t) - \frac{1}{K}\sum_{k=1}^{K}\nabla h_k(x_k^t)\right\|^2 + 2\mathbb{E}\left\|\nabla g_k(x_k^t)\nabla f(\bar{y}^t) - \frac{1}{K}\sum_{k=1}^{K}\nabla g_k(x_k^t)\nabla f(\bar{y}^t)\right]\right\|^2
$$

$$
\stackrel{(c)}{\leq} \underbrace{2\,\mathbb{E}\left\|\nabla h_k(x_k^t) - \frac{1}{K}\sum_{k=1}^{K}\nabla h_k(x_k^t)\right\|^2}_{\text{Term IX}} + \underbrace{2B_f^2\,\mathbb{E}\left\|\nabla g_k(x_k^t) - \frac{1}{K}\sum_{k=1}^{K}\nabla g_k(x_k^t)\right]\right\|^2}_{\text{Term X}}
$$

$$
\stackrel{(d)}{\leq} (6L_h^2 + 6B_f^2 L_g^2)\mathbb{E}\left\|x_k^t - \bar{x}^t\right\|^2 + \frac{6L_h^2 + 6B_f^2 L_g^2}{K}\sum_{k=1}^{K}\mathbb{E}\left\|x_k^t - \bar{x}^t\right\|^2 + 6\Delta_h^2 + 6B_f^2\Delta_g^2
$$

where $(a)$ above uses the definition of $\nabla\Phi_k(x_k^t;\bar{\xi}_k^t)$ in (7) and Assumption 3.3; $(b)$ results from application of Young's inequality; $(c)$ utilized Assumtion 3.2; finally, $(d)$ results from the application of Assumptions 3.2 and 3.4.

Replacing in the upper bound for Term XI, we get

$$
\text{Term XI} \leq 4\mathbb{E}\left\|\frac{1}{K}\sum_{k=1}^{K}\mathbb{E}[\Phi_k(x_k^t;\bar{\xi}_k^t)|\mathcal{F}^t]\right\|^2 + (24L_h^2 + 24B_f^2 L_g^2)\mathbb{E}\left\|x_k^t - \bar{x}^t\right\|^2
$$

$$
+ \frac{24L_h^2 + 24B_f^2 L_g^2}{K}\sum_{k=1}^{K}\mathbb{E}\left\|x_k^t - \bar{x}^t\right\|^2 + \frac{2\sigma_h^2}{|b_h|} + \frac{2\sigma_g^2 B_f^2}{|b_g|} + 24\Delta_h^2 + 24B_f^2\Delta_g^2.
$$

Substituting the bound on Term XI in the bound of Term V, we get

$$
\mathbb{E}\left\|\bar{y}^{t+1} - \frac{1}{K}\sum_{k=1}^{K}g_k(x_k^{t+1})\right\|^2
$$

$$
\leq (1 - \beta^{t+1})^2\,\mathbb{E}\left\|\bar{y}^t - \frac{1}{K}\sum_{k=1}^{K}g_k(x_k^t)\right\|^2 + \frac{8(\eta^t)^2(1-\beta^{t+1})^2 B_g^2}{|b_g|K}\mathbb{E}\left\|\frac{1}{K}\sum_{k=1}^{K}\mathbb{E}[\Phi_k(x_k^t;\bar{\xi}_k^t)|\mathcal{F}^t]\right\|^2
$$

$$
+ \frac{(\eta^t)^2(1-\beta^{t+1})^2 B_g^2(96L_h^2 + 96B_f^2 L_g^2)}{|b_g|K^2}\sum_{k=1}^{K}\mathbb{E}\left\|x_k^t - \bar{x}^t\right\|^2 + \frac{4(\eta^t)^2(1-\beta^{t+1})^2 B_g^2}{|b_h|K}\sigma_h^2
$$

$$
+ \frac{2(\beta^{t+1})^2 + 4(\eta^t)^2(1-\beta^{t+1})^2 B_g^2 B_f^2}{|b_g|K}\sigma_g^2 + \frac{48(\eta^t)^2(1-\beta^{t+1})^2 B_g^2}{|b_g|K}\Delta_h^2 + \frac{48(\eta^t)^2(1-\beta^{t+1})^2 B_f^2 B_g^2}{|b_g|K}\Delta_g^2.
$$

Therefore, we have proof of Lemma. $\qquad\square$

Next, we show descent in the potential function specially designed to show convergence of Algorithm 2. For this purpose, we define the potential function as

$$
V^t = \mathbb{E}[\Phi(\bar{x}^t)] + \mathbb{E}\left\|\bar{y}^t - \frac{1}{K}\sum_{k=1}^{K}g_k(x_k^t)\right\|^2. \tag{21}
$$

Next, we derive the descent in the potential function.

**Lemma F.6** (**Descent in Potential Function**). *Under Assumptions 3.2-3.4 with the choice of momentum-parameter* $\beta^{t+1} = c_\beta \eta^t$ *with* $c_\beta = 4B_g^4 L_f^2$ *where step-size* $\eta^t$ *is chosen such that*

$$
\eta^t \leq \left\{ \frac{|b_g|K}{2(L_\Phi |b_g|K + 8B_g^2)}, \frac{|b_g|K\left(L_h^2 + 2B_f^2 L_g^2 + 4B_g^4 L_f^2\right)}{B_g^2\left(96L_h^2 + 96B_f^2 L_g^2\right)} \right\}
$$

*the iterates generated by Algorithm 2 satisfy*

$$
V^{t+1} - V^t \le -\frac{\eta^t}{2}\mathbb{E}\big\|\nabla\Phi(\bar{x}^t)\big\|^2 + \eta^t\big(2L_h^2 + 4B_f^2 L_g^2 + 8B_g^4 L_F^2\big)\frac{1}{K}\sum_{k=1}^{K}\mathbb{E}\|x_k^t - \bar{x}^t\|^2
$$

$$
+ \frac{2(\eta^t)^2 L_\Phi}{K|b_h|}\sigma_h^2 + \frac{4(\eta^t)^2 B_g^2}{|b_h|K}\sigma_h^2 + \frac{2(\eta^t)^2 L_\Phi B_f^2}{|b_g|K}\sigma_g^2 + \frac{(\eta^t)^2(2c_\beta^2 + 4B_g^2 B_f^2)}{|b_g|K}\sigma_g^2
$$

$$
+ \frac{48(\eta^t)^2 B_g^2}{|b_g|K}\Delta_h^2 + \frac{48(\eta^t)^2 B_f^2 B_g^2}{|b_g|K}\Delta_g^2.
$$

*Proof.* From the definition of $V^t$ in (21) and using Lemmas F.3 and F.5, we get

$$
V^{t+1} - V^t = \mathbb{E}[\Phi(\bar{x}^{t+1}) - \Phi(\bar{x}^t)] + \mathbb{E}\Big\|\bar{y}^{t+1} - \frac{1}{K}\sum_{k=1}^{K}g_k(x_k^{t+1})\Big\|^2 - \mathbb{E}\Big\|\bar{y}^t - \frac{1}{K}\sum_{k=1}^{K}g_k(x_k^t)\Big\|^2
$$

$$
\le -\frac{\eta^t}{2}\mathbb{E}\big\|\nabla\Phi(\bar{x}^t)\big\|^2 - \Big(\frac{\eta^t}{2} - (\eta^t)^2 L_\Phi - \frac{8(\eta^t)^2 B_g^2}{|b_g|K}\Big)\mathbb{E}\Big\|\frac{1}{K}\sum_{k=1}^{K}\mathbb{E}\big[\nabla\Phi_k(x_k^t;\bar{\xi}_k^t)\big|\mathcal{F}^t\big]\Big\|^2
$$

$$
+ \Big(\eta^t\big(L_h^2 + 2B_f^2 L_g^2 + 4B_g^4 L_F^2\big) + \frac{(\eta^t)^2 B_g^2(96L_h^2 + 96B_f^2 L_g^2)}{|b_g|K}\Big)\frac{1}{K}\sum_{k=1}^{K}\mathbb{E}\|x_k^t - \bar{x}^t\|^2
$$

$$
+ \big(4B_g^4 L_f^2\eta^t - \beta^{t+1}\big)\mathbb{E}\Big\|\bar{y}^t - \frac{1}{K}\sum_{k=1}^{K}g_k(x_k^t)\Big\|^2 + \frac{2(\eta^t)^2 L_\Phi}{K|b_h|}\sigma_h^2 + \frac{4(\eta^t)^2 B_g^2}{|b_h|K}\sigma_h^2
$$

$$
+ \frac{2(\eta^t)^2 L_\Phi B_f^2}{|b_g|K}\sigma_g^2 + \frac{2(\beta^{t+1})^2 + 4(\eta^t)^2 B_g^2 B_f^2}{|b_g|K}\sigma_g^2 + \frac{48(\eta^t)^2 B_g^2}{|b_g|K}\Delta_h^2 + \frac{48(\eta^t)^2 B_f^2 B_g^2}{|b_g|K}\Delta_g^2
$$

$$
\overset{(a)}{\le} -\frac{\eta^t}{2}\mathbb{E}\big\|\nabla\Phi(\bar{x}^t)\big\|^2 + \eta^t\big(2L_h^2 + 4B_f^2 L_g^2 + 8B_g^4 L_F^2\big)\frac{1}{K}\sum_{k=1}^{K}\mathbb{E}\|x_k^t - \bar{x}^t\|^2
$$

$$
+ \frac{2(\eta^t)^2 L_\Phi}{K|b_h|}\sigma_h^2 + \frac{4(\eta^t)^2 B_g^2}{|b_h|K}\sigma_h^2 + \frac{2(\eta^t)^2 L_\Phi B_f^2}{|b_g|K}\sigma_g^2 + \frac{(\eta^t)^2(2c_\beta^2 + 4B_g^2 B_f^2)}{|b_g|K}\sigma_g^2
$$

$$
+ \frac{48(\eta^t)^2 B_g^2}{|b_g|K}\Delta_h^2 + \frac{48(\eta^t)^2 B_f^2 B_g^2}{|b_g|K}\Delta_g^2.
$$

where $(a)$ results from the choice of $\beta^t$ and $\eta_t$ given in the statement of the Lemma.

Therefore, we have the proof. $\qquad\square$

**Theorem F.7 (Potential Function).** *Under Assumptions 3.2-3.4 and the choice of step-size $\eta^t = \eta$ such that we have*

$$
\eta \le \frac{1}{3I\big(24L_h^2 + 24B_f^2 L_g^2\big)^{1/2}}
$$

*the iterates generated by Algorithm 2 satisfy*

$$
V^T - V^0 \le -\frac{\eta}{2}\sum_{t=0}^{T-1}\mathbb{E}\big\|\nabla\Phi(\bar{x}^t)\big\|^2 + \eta^3(I-1)^2\frac{\big(10L_h^2 + 20B_f^2 L_g^2 + 40B_g^4 L_F^2\big)}{|b_h|}\sigma_h^2\,T
$$

$$
+ \frac{2\eta^2 L_\Phi}{K|b_h|}\sigma_h^2\,T + \frac{4\eta^2 B_g^2}{|b_h|K}\sigma_h^2\,T + \eta^3(I-1)^2\frac{\big(10B_f^2 L_h^2 + 20B_f^4 L_g^2 + 40B_f^2 B_g^4 L_F^2\big)}{|b_g|}\sigma_g^2\,T
$$

$$
+ \frac{2\eta^2 L_\Phi B_f^2}{|b_g|K}\sigma_g^2\,T + \frac{\eta^2(2c_\beta^2 + 4B_f^2 B_g^2)}{|b_g|K}\sigma_g^2\,T + \eta^3(I-1)^2\big(30L_h^2 + 60B_f^2 L_g^2 + 120B_g^4 L_F^2\big)\Delta_h^2\,T
$$

$$
+ \frac{48\eta^2 B_g^2}{|b_g|K}\Delta_h^2\,T + \eta^3(I-1)^2\big(30B_f^2 L_h^2 + 60B_f^4 L_g^2 + 120B_f^2 B_g^4 L_F^2\big)\Delta_g^2\,T + \frac{48\eta^2 B_f^2 B_g^2}{|b_g|K}\Delta_g^2\,T.
$$

*Proof.* Telescoping the sum of Lemma F.6 for $t = \{0, 1, \ldots, T-1\}$, we get

$$V^T - V^0 \leq -\frac{\eta}{2} \sum_{t=0}^{T-1} \mathbb{E}\|\nabla\Phi(\bar{x}^t)\|^2 + \eta\big(2L_h^2 + 4B_f^2 L_g^2 + 8B_g^4 L_F^2\big) \underbrace{\sum_{t=0}^{T-1} \frac{1}{K} \sum_{k=1}^{K} \mathbb{E}\|x_k^t - \bar{x}^t\|^2}_{\text{Term XIII}}$$

$$+ \frac{2\eta^2 L_\Phi}{K|b_h|}\sigma_h^2\, T + \frac{4\eta^2 B_g^2}{|b_h|K}\sigma_h^2\, T + \frac{2\eta^2 L_\Phi B_f^2}{|b_g|K}\sigma_g^2\, T + \frac{\eta^2(2c_\beta^2 + 4B_g^2 B_f^2)}{|b_g|K}\sigma_g^2\, T$$

$$+ \frac{48\eta^2 B_g^2}{|b_g|K}\Delta_h^2\, T + \frac{48\eta^2 B_f^2 B_g^2}{|b_g|K}\Delta_g^2\, T. \tag{22}$$

We bound Term XIII in (22) using Lemma (F.4). Note that we have from Lemma (F.4)

$$\frac{1}{K}\sum_{k=1}^{K}\mathbb{E}\|x_k^t - \bar{x}^t\|^2 \leq (I-1)\big(24L_h^2 + 24B_f^2 L_g^2\big)\sum_{\ell=t_s}^{t-1}\frac{(\eta^\ell)^2}{K}\sum_{k=1}^{K}\mathbb{E}\|x_k^\ell - \bar{x}^\ell\|^2$$

$$+ (I-1)\left(\frac{4}{|b_h^t|}\sigma_h^2 + \frac{4B_f^2}{|b_g^t|}\sigma_g^2\right)\sum_{\ell=t_s}^{t-1}(\eta^\ell)^2 + (I-1)\big(12\Delta_h^2 + 12B_f^2\Delta_g^2\big)\sum_{\ell=t_s}^{t-1}(\eta^\ell)^2$$

Summing the above from $t = t_s$ to $t_{s+1} - 1$, we get

$$\sum_{t=t_s}^{t_{s+1}-1}\frac{1}{K}\sum_{k=1}^{K}\mathbb{E}\|x_k^t - \bar{x}^t\|^2 \overset{(a)}{\leq} \eta^2(I-1)\big(24L_h^2 + 24B_f^2 L_g^2\big)\sum_{t=t_s}^{t_{s+1}-1}\sum_{\ell=t_s}^{t-1}\frac{1}{K}\sum_{k=1}^{K}\mathbb{E}\|x_k^\ell - \bar{x}^\ell\|^2$$

$$+ \eta^2(I-1)^2 I\left(\frac{4}{|b_h^t|}\sigma_h^2 + \frac{4B_f^2}{|b_g^t|}\sigma_g^2\right) + \eta^2(I-1)^2 I\big(12\Delta_h^2 + 12B_f^2\Delta_g^2\big)$$

$$\overset{(b)}{\leq} \eta^2(I-1)\big(24L_h^2 + 24B_f^2 L_g^2\big)\sum_{t=t_s}^{t_{s+1}-1}\sum_{\ell=t_s}^{t_{s+1}-1}\frac{1}{K}\sum_{k=1}^{K}\mathbb{E}\|x_k^\ell - \bar{x}^\ell\|^2$$

$$+ \eta^2(I-1)^2 I\left(\frac{4}{|b_h^t|}\sigma_h^2 + \frac{4B_f^2}{|b_g^t|}\sigma_g^2\right) + \eta^2(I-1)^2 I\big(12\Delta_h^2 + 12B_f^2\Delta_g^2\big)$$

$$\overset{(c)}{\leq} \eta^2(I-1) I\big(24L_h^2 + 24B_f^2 L_g^2\big)\sum_{t=t_s}^{t_{s+1}-1}\frac{1}{K}\sum_{k=1}^{K}\mathbb{E}\|x_k^t - \bar{x}^t\|^2$$

$$+ \eta^2(I-1)^2 I\left(\frac{4}{|b_h^t|}\sigma_h^2 + \frac{4B_f^2}{|b_g^t|}\sigma_g^2\right) + \eta^2(I-1)^2 I\big(12\Delta_h^2 + 12B_f^2\Delta_g^2\big)$$

where in $(a)$ we have used the fact that $\eta^t = \eta$ for all $t \in [T]$ and $(t-1) - t_s \leq I - 1$ for $t \in [t_s, t_{s+1} - 1]$; $(b)$ results from the fact that $t \leq t_{s+1}$; finally, $(c)$ again uses the fact that $(t-1) - t_s \leq I - 1$ for $t \in [t_s, t_{s+1} - 1]$.

Summing the above from $s = \{0, 1, \ldots, S\}$ and using the fact that $S \times I = T - 1$, we get

$$\sum_{t=0}^{T-1}\frac{1}{K}\sum_{k=1}^{K}\mathbb{E}\|x_k^t - \bar{x}^t\|^2 \leq \eta^2 I^2\big(24L_h^2 + 24B_f^2 L_g^2\big)\sum_{t=0}^{T-1}\frac{1}{K}\sum_{k=1}^{K}\mathbb{E}\|x_k^t - \bar{x}^t\|^2$$

$$+ \eta^2(I-1)^2\left(\frac{4}{|b_h^t|}\sigma_h^2 + \frac{4B_f^2}{|b_g^t|}\sigma_g^2\right) T + \eta^2(I-1)^2\big(12\Delta_h^2 + 12B_f^2\Delta_g^2\big) T.$$

Rearranging the terms, we get

$$\Big(1 - \eta^2 I^2\big(24L_h^2 + 24B_f^2 L_g^2\big)\Big)\sum_{t=0}^{T-1}\frac{1}{K}\sum_{k=1}^{K}\mathbb{E}\|x_k^t - \bar{x}^t\|^2 \leq \eta^2(I-1)^2\left(\frac{4}{|b_h^t|}\sigma_h^2 + \frac{4B_f^2}{|b_g^t|}\sigma_g^2\right) T$$

$$+ \eta^2(I-1)^2\big(12\Delta_h^2 + 12B_f^2\Delta_g^2\big) T.$$

Finally, choosing $\eta \leq \frac{1}{3I\left(24L_h^2 + 24B_f^2 L_g^2\right)^{1/2}}$, such that we have $1 - \eta^2 I^2\left(24L_h^2 + 24B_f^2 L_g^2\right) \geq 8/9$, utilizing this we get

$$\text{Term XIII} := \sum_{t=0}^{T-1} \frac{1}{K} \sum_{k=1}^{K} \mathbb{E}\|x_k^t - \bar{x}^t\|^2$$

$$\leq \eta^2 (I-1)^2 \left(\frac{5}{|b_h^t|}\sigma_h^2 + \frac{5B_f^2}{|b_g^t|}\sigma_g^2\right) T + \eta^2 (I-1)^2\left(15\Delta_h^2 + 15B_f^2 \Delta_g^2\right) T.$$

Finally, substituting the bound on Term XIII in (22), we get

$$V^T - V^0 \leq -\frac{\eta}{2}\sum_{t=0}^{T-1}\mathbb{E}\big\|\nabla\Phi(\bar{x}^t)\big\|^2 + \eta^3 (I-1)^2 \frac{\left(10L_h^2 + 20B_f^2 L_g^2 + 40B_g^4 L_F^2\right)}{|b_h|}\sigma_h^2\, T$$

$$+ \frac{2\eta^2 L_\Phi}{K|b_h|}\sigma_h^2\, T + \frac{4\eta^2 B_g^2}{|b_h|K}\sigma_h^2\, T + \eta^3 (I-1)^2 \frac{\left(10B_f^2 L_h^2 + 20B_f^4 L_g^2 + 40B_f^2 B_g^4 L_F^2\right)}{|b_g|}\sigma_g^2\, T$$

$$+ \frac{2\eta^2 L_\Phi B_f^2}{|b_g|K}\sigma_g^2\, T + \frac{\eta^2(2c_\beta^2 + 4B_f^2 B_g^2)}{|b_g|K}\sigma_g^2\, T + \eta^3 (I-1)^2\left(30L_h^2 + 60B_f^2 L_g^2 + 120B_g^4 L_F^2\right)\Delta_h^2\, T$$

$$+ \frac{48\eta^2 B_g^2}{|b_g|K}\Delta_h^2\, T + \eta^3 (I-1)^2\left(30B_f^2 L_h^2 + 60B_f^4 L_g^2 + 120B_f^2 B_g^4 L_F^2\right)\Delta_g^2\, T + \frac{48\eta^2 B_f^2 B_g^2}{|b_g|K}\Delta_g^2\, T.$$

Therefore, we have the proof. $\qquad\square$

Now, we are finally ready to prove Theorem 4.3.

*Proof.* Assuming $|b_h| = |b_g| = |b|$ and defining $\bar{L}_{f,g} := 10L_h^2 + B_f^2 L_g^2 + 40B_g^4 L_f^2$. Rearranging the terms in the expression of Theorem F.7 and multiplying both sides by $2/\eta T$ we get

$$\frac{1}{T}\sum_{t=0}^{T-1}\mathbb{E}\big\|\nabla\Phi(\bar{x}^t)\big\|^2 \leq \frac{2\left[\Phi(\bar{x}^0) - \Phi(x^*) + \big\|\bar{y}^0 - g(\bar{x}^0)\big\|^2\right]}{\eta T} + \eta^2 (I-1)^2\left[\frac{2\bar{L}_{f,g}}{|b|}\sigma_h^2 + \frac{2B_f^2 \bar{L}_{f,g}}{|b|}\sigma_g^2\right]$$

$$+ \eta^2 (I-1)^2\left[6\bar{L}_{f,g}\Delta_h^2 + 6B_f^2 \bar{L}_{f,g}\Delta_g^2\right] + \eta\left[\frac{4L_\Phi + 8B_g^2}{|b|K}\sigma_h^2 + \frac{4L_\Phi B_f^2 + 4c_\beta^2 + 8B_f^2 B_g^2}{|b|K}\sigma_g^2\right]$$

$$+ \eta\left[\frac{96B_g^2}{|b|K}\Delta_h^2 + \frac{96B_f^2 B_g^2}{|b|K}\Delta_g^2\right],$$

where the first term on the right follows from the fact that $\Phi(\bar{x}^T) \geq \Phi(x^*)$ and $\|\bar{y}^T - 1/K\sum_{k=1}^{K} g_k(x_k^T)\|^2 \geq 0$.

Next, choosing $\eta = \sqrt{\frac{|b|K}{T}}$ then for $T \geq \left(216L_h^2 + 216B_f^2 L_g^2\right)I^2|b|K$ such that $\eta \leq \frac{1}{3I\left(24L_h^2 + 24B_f^2 L_g^2\right)^{1/2}}$ in Theorem F.7 is satisfied, we get the following

$$\frac{1}{T}\sum_{t=0}^{T-1}\mathbb{E}\big\|\nabla\Phi(\bar{x}^t)\big\|^2 \leq \frac{2\left[\Phi(\bar{x}^0) - \Phi(x^*) + \big\|\bar{y}^0 - g(\bar{x}^0)\big\|^2\right]}{\sqrt{|b|KT}} + \frac{K(I-1)^2}{T}\left[2\bar{L}_{f,g}\sigma_h^2 + 2B_f^2 \bar{L}_{f,g}\sigma_g^2\right]$$

$$+ \frac{|b|K(I-1)^2}{T}\left[6\bar{L}_{f,g}\Delta_h^2 + 6B_f^2 \bar{L}_{f,g}\Delta_g^2\right] + \frac{1}{\sqrt{|b|KT}}\left[\left(4L_\Phi + 8B_g^2\right)\sigma_h^2 + \left(4L_\Phi B_f^2 + 4c_\beta^2 + 8B_f^2 B_g^2\right)\sigma_g^2\right]$$

$$+ \frac{1}{\sqrt{|b|KT}}\left[96B_g^2\, \Delta_h^2 + 96B_f^2 B_g^2\, \Delta_g^2\right],$$

Explicitly choosing $I = T^{1/4}/(|b|K)^{3/4}$, we get

$$\mathbb{E}\big\|\nabla\Phi(\bar{x}^{a(T)})\big\|^2 \leq \frac{2\left[\Phi(\bar{x}^0) - \Phi(x^*) + \big\|\bar{y}^0 - g(\bar{x}^0)\big\|^2\right]}{\sqrt{|b|KT}} + \frac{C_{\sigma_h}}{\sqrt{|b|KT}}\sigma_h^2 + \frac{C_{\sigma_g}}{\sqrt{|b|KT}}\sigma_g^2$$

$$+ \frac{C_{\Delta_h}}{\sqrt{|b|KT}}\Delta_h^2 + \frac{C_{\Delta_g}}{\sqrt{|b|KT}}\Delta_g^2.$$

where the constants $C_{\sigma_h}$, $C_{\sigma_g}$, $C_{\Delta_f}$, and $C_{\Delta_g}$ are defined as:

$$C_{\sigma_h} = 2\bar{L}_{f,g} + 4L_\Phi + 8B_g^2$$
$$C_{\sigma_g} = 2B_f^2\bar{L}_{f,g} + 4L_\Phi B_f^2 + 4c_\beta^2 + 8B_f^2 B_g^2$$
$$C_{\Delta_f} = 6\bar{L}_{f,g} + 96B_g^2$$
$$C_{\Delta_g} = 6B_f^2\bar{L}_{f,g} + 96B_f^2 B_g^2.$$

The constant $c_\beta$ is defined in the statement of Lemma F.6.

Hence, Theorem 4.3 is proved. $\qquad\qquad\square$

## G   PROOF OF THEOREM 5.2

Let us restate Theorem 5.2 for convenience.

**Theorem G.1.** *For Algorithm 3, choosing the local step-sizes $\eta^t = \eta = \mathcal{O}(\sqrt{I/T})$ and the momentum parameter $\beta = c_\beta\eta$ for all $t \in \{0, 1, \ldots, T-1\}$. Choosing the server step-sizes $\gamma_x = \mathcal{O}(\sqrt{K/T})$ and $\gamma_y = c_{\gamma_y}\gamma_x$. Then under Assumptions 3.2, 3.3, 3.4, and 5.1 for $\bar{x}^{a(T)}$ chosen According to Algorithm 3, we have*

$$\mathbb{E}\big\|\nabla\Phi(\bar{x}^{a(T)})\big\|^2 \leq \mathcal{C}_{Sync}\mathcal{O}\Big(\sqrt{\frac{1}{KT}}\Big) + \mathcal{C}_{Drift}\mathcal{O}\Big(\frac{1}{T}\Big),$$

*for some constants $c_\beta$, $c_{\gamma_y}$, $\mathcal{C}_{Sync}$ and $\mathcal{C}_{Drift}$.*

We note that the proof of Theorem G.1 although different will follow the structure and the steps of the proof of (Yang et al., 2024), therefore, we omit the detailed proofs. Let us first state the main lemmas utilized in the proof of the theorem.

**Lemma G.2** (**Descent in Function Value**). *Under Assumptions 3.2-5.1, the iterates generated by Algorithm 3 satisfy*

$$\mathbb{E}[\Phi(x^{\tau+1}) - \Phi(x^\tau)] \leq -\frac{\gamma_x\eta I}{2}\mathbb{E}\|\nabla\Phi(x^\tau)\|^2$$

$$-\left[\frac{\gamma_x\eta I}{2} - \frac{L_\Phi^2\gamma_x^2\eta^2 I^2}{2}\right]\mathbb{E}\Big\|\frac{1}{K}\sum_{k=1}^{K}\frac{1}{I}\sum_{t=\tau I}^{(\tau+1)I-1}\mathbb{E}[\nabla\Phi_k(x_k^t;\bar{\xi}_k^t)|\mathcal{G}_t]\Big\|$$

$$+\frac{\gamma_x^2\eta^2 I^2 L_\Phi^2}{2}\left[\frac{\sigma_h^2 + B_f^2\sigma_g^2}{KI}\right]$$

$$+\gamma_x\eta I\big[L_h^2 + 2B_f^2 L_g^2 + 16B_g^4 L_f^2\big]\frac{1}{K}\sum_{k=1}^{K}\frac{1}{I}\sum_{t=\tau I}^{(\tau+1)I-1}\mathbb{E}\|x_k^{t+1} - x^\tau\|^2$$

$$+\gamma_x\eta I\big[12B_g^2 L_f^2\big]\frac{1}{K}\sum_{k=1}^{K}\frac{1}{I}\sum_{t=\tau I}^{(\tau+1)I-1}\mathbb{E}\|y_k^t - y^\tau\|^2$$

$$+\gamma_x\eta I\big[12B_g^2 L_f^2\big]\mathbb{E}\|y^\tau - g(x^\tau)\|^2.$$

**Lemma G.3** (**Drift in $y$-Updates**). *Under Assumptions 3.2-5.1, the iterates generated by Algorithm 3 satisfy*

$$\frac{1}{K}\sum_{k=1}^{K}\frac{1}{I}\sum_{t=\tau I}^{(\tau+1)I-1}\mathbb{E}\|y_k^t - y^\tau\|^2 \leq \frac{\beta^2 I}{1 - 4\beta^2 I}\Big[\sigma_g^2 + 4\Delta_g^2 + 4\eta^2 I B_g^2\sigma_X^2 + 4\mathbb{E}\|y^\tau - g(x^\tau)\|^2\Big],$$

*where $\sigma_X^2$ is defined as $\sigma_X^2 := 3\sigma_h^2 + 3B_f^2\sigma_g^2 + 6B_h^2 + 6B_g^2 B_f^2$.*

**Lemma G.4** (**Drift in $x$-Updates**). *Under Assumptions 3.2-5.1, the iterates generated by Algorithm 3 satisfy*

$$\frac{1}{K}\sum_{k=1}^{K}\frac{1}{I}\sum_{t=\tau I}^{(\tau+1)I-1}\mathbb{E}\|x_k^{t+1} - x^\tau\|^2 \leq \eta^2 I\sigma_X^2.$$

*Similarly, we bound the term*

$$\mathbb{E}\|x^{\tau+1} - x^\tau\|^2 \leq \gamma_x^2\eta^2 I\sigma_X^2.$$

**Lemma G.5** (**Descent in the estimate of** $y$). *Under Assumptions 3.2-5.1, the iterates generated by Algorithm 3 satisfy*

$$\mathbb{E}\|y^{\tau+1} - g(x^{\tau+1})\|^2 - \mathbb{E}\|y^\tau - g(x^\tau)\|^2 \leq \left[\delta + 6I^2\gamma_y^2\beta^2 - \gamma_y\beta I\right]\mathbb{E}\|y^\tau - g(x^\tau)\|$$

$$+ \left[\frac{6\gamma_y^2\beta^2 I^2}{K} + 8\gamma_y\beta I\right]\left[\frac{1}{K}\sum_{k=1}^{K}\frac{1}{I}\sum_{t=\tau I}^{(\tau+1)I-1}\mathbb{E}\|y_k^t - y^\tau\|^2 + B_g^2\mathbb{E}\|x_k^{t+1} - x^\tau\|^2\right]$$

$$+ B_g^2\mathbb{E}\|x^{\tau+1} - x^\tau\|^2 + \left[\frac{4B_g^2}{\delta_1} + \frac{L_g}{2}\right]\sigma_X^2\gamma_x^2\eta^2 I,$$

*where $\delta := \delta_1 + \frac{\eta^2\gamma_x^2 I L_g}{2}\sigma_X^2$ and $\delta_1 > 0$ is a parameter to be chosen later.*

Next, we design the potential function as

$$V_\tau = \mathbb{E}[\Phi(x^\tau) + \|y^\tau - g(x^\tau)\|^2],$$

and our goal is to analyze the descent in the potential function. We analyze the term

$$V_{\tau+1} - V_\tau = \mathbb{E}[\Phi(x^{\tau+1}) - \Phi(x^\tau)] + \mathbb{E}[\|y^{\tau+1} - g(x^{\tau+1})\|^2 - \|y^\tau - g(x^\tau)\|^2] \quad (23)$$

Choosing the learning rates such that, we have

$$\delta_1 = B_g^2 L_f^2 \gamma_x\eta I, \gamma_y\beta \leq \frac{8}{6KI}, \beta \leq \frac{3}{64}\gamma_y, \beta \leq \frac{1}{\sqrt{8I}}, \eta\gamma_x \leq \frac{2B_g^2 L_f^2}{L_g\sigma_X^2},$$

$$\delta \leq 2B_g^2 L_f^2 \gamma_x\eta I, \gamma_y^2\beta^2 \leq \frac{2b_g^2 l_f^2 \gamma_x\eta}{I}, \gamma_y\beta \leq 28B_g^2 L_f^2 \gamma_x\eta, \gamma_x\eta \leq \frac{1}{L_\Phi^2 I}$$

This choice of parameters implies that we will have

$$V_{\tau+1} - V_\tau = \mathbb{E}[\Phi(x^{\tau+1}) - \Phi(x^\tau)] + \mathbb{E}[\|y^{\tau+1} - g(x^{\tau+1})\|^2 - \|y^\tau - g(x^\tau)\|^2]$$

$$\leq -\frac{\gamma_x\eta I}{2}\mathbb{E}\|\nabla\Phi(x^\tau)\|^2$$

$$+ \frac{\gamma_x^2\eta^2 I^2 L_\Phi^2}{2}\left[\frac{\sigma_h^2 + B_f^2\sigma_g^2}{KI}\right]$$

$$+ \gamma_x\eta I\left[L_h^2 + 2B_f^2 L_g^2 + 16B_g^4 L_f^2\right] \times \eta^2 I\sigma_X^2$$

$$+ \gamma_x\eta I\left[12B_g^2 L_f^2\right] \times 2\beta^2 I\left[\sigma_g^2 + 4\Delta_g^2 + 4\eta^2 IB_g^2\sigma_X^2\right]$$

$$+ \left[\frac{6\gamma_y^2\beta^2 I^2}{K} + 8\gamma_y\beta I\right]2\beta^2 I\left[\sigma_g^2 + 4\Delta_g^2 + 4\eta^2 IB_g^2\sigma_X^2\right]$$

$$+ \left[\frac{6\gamma_y^2\beta^2 I^2}{K} + 8\gamma_y\beta I\right]B_g^2\eta^2 I\sigma_X^2$$

$$+ B_g^2\gamma_x^2\eta^2 I\sigma_X^2 + \left[\frac{4B_g^2}{\delta_1} + \frac{L_g}{2}\right]\sigma_X^2\gamma_x^2\eta^2 I,$$

Finally, rearranging the terms and multiplying both sides by $\frac{2}{\gamma_x\eta I}$, telescoping the sum, and choosing the step-sizes such that we have

$$\gamma_x = \mathcal{O}\left(\sqrt{\frac{K}{T}}\right), \gamma_y = \mathcal{O}(\gamma_x)$$

$$\eta = \mathcal{O}\left(\frac{\sqrt{I}}{\sqrt{T}}\right), \beta = \mathcal{O}\left(\frac{\sqrt{I}}{\sqrt{T}}\right)$$

This yields the statement of the theorem. $\qquad\square$

