# OpenReview forum: "FEDERATED COMPOSITIONAL OPTIMIZATION: THE IMPACT OF TWO-SIDED LEARNING RATES ON COMMUNICATION EFFICIENCY"
_ICLR.cc/2025/Conference — ICLR 2025 Conference Withdrawn Submission_

### Official Review · Reviewer_c7GE · 2024-11-01

**Soundness:** 2
**Presentation:** 3
**Contribution:** 2
**Rating:** 3
**Confidence:** 4

**Summary:**

This paper studies federated learning methods for distributionally robust optimization problems. It proposes two algorithms and provides convergence analysis. The experimental results validate their performance.

**Strengths:**

1. This paper is well written. The motivation and limitation of existing methods are well stated.

2. The proof of Theorem 4.1 is interesting.

**Weaknesses:**

1. Table 1 has many errors. ComFedL and FedNest communicate in every iteration. Their communication complexity is $O(\epsilon^{-2})$. The communication complexity of Local-SCGDM is $O(\epsilon^{-1.5})$, not $O(\epsilon^{-1})$. For FedBiO, its computation complexity is $O(\epsilon^{-1.5})$,  not $O(\epsilon^{-2.5})$, and its communication complexity is $O(\epsilon^{-1})$. In addition, FedNest needs to take into account the inner loop length.

2. This paper states that DS-FedDRO requires stronger assumptions than FedDRO. However, it seems they use the same assumptions, as Assumption 5.1 is same as Assumption 3.4. Why does DS-FedDRO have a better communication complexity? It would be good to provide detailed discussions about this improvement.

3. To estimate $g$, why do DS-FedDRO and FedDRO use different methods?

4. Some claims in this paper are incorrect. For example, this paper states that  this is the first work that ensures linear speed up in a federated CO setting. Actually, existing federated bilevel optimization algorithms, e.g., FedBiO and FedMBO in Table 1,  have achieved linear speedup. In addition, existing federated compositional optimization algorithms [1,2] also have achieved linear speedup.

5. The novelty is incremental. A key contribution of this paper is to communicate $y$. However, existing federated compositional methods, such as Local-SCGDM, have already proposed to communicate it. Moreover, as a special bilevel optimization problem, existing federated bilevel optimization algorithms, such as FedBiO, also communicate the inner level function. Due to these existing methods, I don't think communicating $y$ is the contribution of this paper.  For the convergence analysis, I didn't see new challenges in this paper compared to those existing federated compositional and bilevel optimization problems. It might be good if the authors could point out why those methods do not work for FedDRO problems.

6. It seems that Algorithm 2 requires to communicate $y$ in every iteration. Then, the communication complexity in Corollary 4.4 is incorrect. It should be same as the number of iterations.


[1] Wu, Xidong, et al. "Federated conditional stochastic optimization." Advances in Neural Information Processing Systems 36 (2024).

[2] Gao, Hongchang. "A Doubly Recursive Stochastic Compositional Gradient Descent Method for Federated Multi-Level Compositional Optimization." Forty-first International Conference on Machine Learning. 2024.

**Questions:**

Please see Weaknesses.

---

### Official Review · Reviewer_2rho · 2024-11-03

**Soundness:** 3
**Presentation:** 2
**Contribution:** 2
**Rating:** 5
**Confidence:** 3

**Summary:**

This work addresses challenges in compositional optimization (CO) within federated learning (FL), a framework essential for applications like distributionally robust optimization (DRO), meta-learning, and reinforcement learning. Existing methods for CO often require impractically large batch gradients or are inefficient in communication, making them unsuitable for distributed FL environments with heterogeneous data.

To overcome these issues, the authors propose two FedAvg-type algorithms specifically for non-convex CO in FL settings. First, they show that the standard FedAvg approach struggles with federated CO problems, as data heterogeneity amplifies bias in local gradient estimates. To manage this bias, they suggest either increased communication or the use of two-sided learning rates.

1. **FedDRO**: This algorithm leverages the CO structure to create a communication-efficient strategy for FedAvg that controls bias in compositional gradient estimation, achieving \(\mathcal{O}(\epsilon^{-2})\) sample and \(\mathcal{O}(\epsilon^{-3/2})\) communication complexity.

2. **DS-FedDRO**: This two-sided learning rate algorithm eliminates the need for extra communication, achieving optimal sample complexity of \(\mathcal{O}(\epsilon^{-2})\) and communication complexity of \(\mathcal{O}(\epsilon^{-1})\), demonstrating the effectiveness of two-sided learning rates in federated CO.

Both algorithms avoid the need for large batch gradients and provide linear speedup with the number of clients. The authors validate their theoretical results with empirical studies on large-scale DRO tasks.

**Strengths:**

This paper is clear, well-organized, and effectively presents its contributions.

The introduction is particularly strong, providing a solid overview of the motivation and challenges in compositional optimization (CO) within federated learning (FL). The authors clearly articulate the need for more efficient algorithms due to the distributed nature of data in FL and the unique challenges (C1-C3) posed by the compositional structure of the objectives. The contributions are also well-defined, with items providing a concise and informative summary of the paper’s primary advances.

The problem formulation is thorough and establishes a clear foundation for the study. The authors present the formalism in a way that is both rigorous and accessible, allowing readers to quickly understand the specifics of the CO problem and how it fits into the FL context.

Additionally, the examples demonstrating how CO can be used to reformulate distributionally robust optimization (DRO) problems are insightful and help ground the theoretical concepts in practical applications. These examples make the technical aspects more tangible and illustrate how the proposed methods could be applied in real-world settings.

Table 1 is a valuable addition, as it provides a comparative overview of the theoretical results, enabling readers to quickly see the advantages of the proposed methods relative to existing approaches. The assumptions, definitions, lemmas, and theorems throughout the paper are clearly stated and rigorously justified, reflecting a high standard of technical precision.

Algorithm 2, in particular, is well-detailed and clearly explained, making it easy to follow the logic and mechanics behind the proposed solution. The authors’ step-by-step presentation ensures that the algorithm’s strengths and innovations are apparent to readers.

Finally, the experimental section includes results for a ResNet20 model, which adds an empirical dimension to the theoretical findings. These experiments help validate the proposed algorithms and demonstrate their practical effectiveness on a well-known neural network model, further reinforcing the relevance of the work.

**Weaknesses:**

1. **Abstract**: While the abstract effectively conveys the main ideas, it is somewhat lengthy and could benefit from a more concise presentation. A shorter abstract would make the paper's contributions easier to grasp at a glance.

2. **Literature Review**: The literature review is somewhat limited and does not discuss key papers that have addressed two-sided (server-side and client-side) step sizes, which are relevant for federated learning. Notable works in this area include:
    - Malinovsky, Grigory, Konstantin Mishchenko, and Peter Richtárik. "Server-side stepsizes and sampling without replacement provably help in federated optimization." Proceedings of the 4th International Workshop on Distributed Machine Learning. 2023.
    - Charles, Zachary, and Jakub Konečný. "On the outsized importance of learning rates in local update methods." arXiv preprint arXiv:2007.00878 (2020).
    - Mai, Van Sy, et al. "Federated learning with server learning for non-IID data." 2023 57th Annual Conference on Information Sciences and Systems (CISS). IEEE, 2023.

   These studies employed different step sizes for the server and client to address the effects of heterogeneity. Since these papers are directly related to the concept of controlling client-server divergence, it would be beneficial to include and discuss them within the context of this work.

3. **Client Drift Reduction in Federated Learning**: In federated learning, several methods like SCAFFOLD, ProxSkip, and FedLin aim to reduce the impact of client drift by learning specific shifts that adjust during training. This approach is a widely recognized strategy to mitigate heterogeneity effects. However, the current paper does not provide a detailed discussion of this aspect. Could you consider addressing how your approach compares to client drift reduction methods?
    - Karimireddy, Sai Praneeth, et al. "Scaffold: Stochastic controlled averaging for federated learning." International conference on machine learning. PMLR, 2020.
    - Mishchenko, Konstantin, et al. "Proxskip: Yes! local gradient steps provably lead to communication acceleration! finally!." International Conference on Machine Learning. PMLR, 2022.
    - Mitra, Aritra, et al. "Linear convergence in federated learning: Tackling client heterogeneity and sparse gradients." Advances in Neural Information Processing Systems 34 (2021): 14606-14619.

4. **Complexity Analysis**: The stated complexities are expressed in Big O notation with respect to parameters \(\varepsilon\), \(K\), and \(m\). However, other important constants, such as the Lipschitz constants, are not included in the complexity bounds. Including these parameters would provide a more complete picture of the algorithm's theoretical performance.

5. **Generality of Results**: The results presented are for the general non-convex case, which are understandably conservative. However, the paper does not explore results for strongly convex or convex cases, which could potentially yield improved complexity bounds. Furthermore, even for the non-convex case, there are no results under the PL (Polyak-Łojasiewicz) condition. Could you comment on whether obtaining such results would be more challenging in this setting?

6. **Experimental Scope**: The experiments are conducted on a ResNet model using the CIFAR-10 and CIFAR-100 datasets. Although these provide some insights, the scope of datasets and models could be broadened to present a more extensive evaluation.

7. **Controllable Experimental Setting**: The current experiments focus on neural networks, but they do not include results for a controlled setting where assumptions like smoothness and heterogeneity are explicitly satisfied (e.g., logistic regression with non-convex regularization). It would be helpful to add results for such a controlled setting to validate theoretical assumptions more concretely.

8. **Convergence Plots**: The experimental results currently include only accuracy plots. Since the optimization theory pertains to convergence, it would be useful to add loss convergence plots to illustrate the optimization performance directly.

9. **Assumptions 3.4 and 5.2**: The difference between Assumptions 3.4 and 5.2, which both relate to bounded heterogeneity, is unclear. To improve clarity, consider giving these assumptions distinct names and providing a brief discussion that highlights their specific roles in the analysis.

**Questions:**

Please check Weaknesses section.

---

### Official Review · Reviewer_p6KL · 2024-11-04

**Soundness:** 3
**Presentation:** 3
**Contribution:** 3
**Rating:** 6
**Confidence:** 2

**Summary:**

This paper addresses compositional optimization (CO) within federated learning (FL), highlighting the limitations of existing methods that rely on large batch gradients or exhibit suboptimal communication efficiency due to data heterogeneity. The authors propose two efficient FedAvg-type algorithms: FedDRO, which leverages the compositional structure to control bias in gradient estimation, achieving O(ϵ−2) sample and O(ϵ−3/2) communication complexity; and DS-FedDRO, a two-sided learning rate algorithm that eliminates the need for extra communication while achieving optimal O(ϵ−2) sample and O(ϵ−1) communication complexity. Both algorithms improve efficiency without requiring large batch gradients and demonstrate linear speedup with the number of clients.

**Strengths:**

1 The improvement process in the methodology section is very clear. It first establishes the incapability of vanilla FedAvg to solve compositional optimization (CO) problems. Based on the identified issues, the authors propose sharing $y_k$ in each iteration and modifying the classical FedAvg aggregation to implement a two-sided update, where the server incrementally updates both x and y. This leads to the introduction of the two proposed algorithms.

2 One of the key advantages of DS-FedDRO is its ability to achieve linear speedup while significantly improving communication performance compared to existing method. The communication complexity of O(ϵ−1) matches the best-known performance for standard federated learning problems. Additionally, the update rules used in DS-FedDRO and FedDRO are simpler than those in bilevel optimization algorithms, requiring the sharing of fewer sequences.

**Weaknesses:**

I am not very familiar with DRO experiments and would like to ask how the task of classification with an imbalanced dataset is formulated as a DRO problem. Additionally, I noticed that the baseline for this experiment is not included in Table 1. What are the characteristics of the problem for this task?

**Questions:**

Please see weakness.

---

### Note · Authors · 2024-11-21

I have read and agree with the venue's withdrawal policy on behalf of myself and my co-authors.